



# Efficient Bayesian inference for large chaotic dynamical systems

Sebastian Springer[1], Heikki Haario[1,2], Jouni Susiluoto[1,3], Aleksandr Bibov[1,5],
Andrew Davis[3,4], and Youssef Marzouk[3]

[1]Department of Computational and Process Engineering, Lappeenranta University of Technology,
Lappeenranta, FI
[2]Finnish Meteorological Institute, Helsinki, Finland
[3]Department of Aeronautics and Astronautics, Massachusetts Institute of Technology, Cambridge, MA
[4]Courant Institute of Mathematical Sciences, New York University, New York, NY
[5]Varjo Technologies Oy, Helsinki, FI

**Correspondence:** Sebastian Springer (sebastian.springer@lut.fi)

**Abstract.**

Estimating parameters of chaotic geophysical models is challenging due to these models' inherent unpredictability. With temporally sparse long-range observations, these models cannot be calibrated using standard least squares or filtering methods. Obvious remedies, such as averaging over temporal and spa-

tial data to characterize the mean behavior, do not capture the subtleties of the underlying dynamics. We perform Bayesian inference of parameters in high-dimensional and computationally demanding chaotic dynamical systems by combining two approaches: (i) measuring model-data mismatch by comparing chaotic attractors, and (ii) mitigating the computational cost of inference by using surrogate models. Specifically, we construct a likelihood function suited to chaotic models by evaluating a distribution over *distances* be-

tween points in the phase space; this distribution defines a summary statistic that depends on the attractor's geometry, rather than on pointwise matching of trajectories. This statistic is computationally expensive to simulate, compounding the usual challenges of Bayesian computation with physical models. Thus we develop an inexpensive surrogate for the log-likelihood via local approximation Markov chain Monte Carlo, which in our simulations reduces the time required for accurate inference by orders of magnitude. We





investigate the behavior of the resulting algorithm on model problems, and then use a quasi-geostrophic model to demonstrate its large-scale application.

# 1 Introduction

Parameter estimation in chaotic models is an important problem for a range of geophysical applications,
with weather prediction and climate modelling the most typical examples. This paper focuses on Bayesian approaches to parameter inference in these settings. Given observational data, the parameters of a dynamical system model are typically inferred by minimizing a cost function that captures model-observation mismatch. In the Bayesian setting, a statistical characterization of this mismatch yields a likelihood function, which enables maximum likelihood estimation or fully Bayesian inference. The latter is typically realized
via Markov chain Monte Carlo (MCMC) methods (Robert and Casella, 2004). Yet common mismatch functions (for instance, the squared Euclidean distance between model outputs and data) are inadequate for chaotic models where small changes in parameters, or even in the tolerances used for numerical solvers, can lead to arbitrarily large differences in model outputs at a given time (Rougier, 2013). We therefore adapt the mismatch function proposed in Haario et al. (2015) to define a likelihood that can be used in
point estimation or Bayesian inference. Our focus here is on the Bayesian approach, which offers a natural means of uncertainty quantification in the parameters. Characterizing the posterior distribution associated with the proposed Bayesian formulation, however, can be computationally prohibitive. Thus we combine our new likelihood with the *local approximation MCMC* methodology introduced in Conrad et al. (2016, 2018); Davis (2018); Davis et al. (2020), which constructs and iteratively refines a surrogate model for the
posterior density during MCMC sampling.

Now we provide some background relevant to our proposed approach. Traditional methods for estimating parameters in chaotic models constrain the problem to shorter time intervals, avoiding the eventual divergence of nearby orbits due to the intrinsic dynamics of the system. A classical example is variational



data assimilation for weather prediction, where the initial states of the model are estimated using observa-
tional data and algorithms such as 4DVar, after which a short-time forecast can be simulated (Asch et al.,
2016).

Sequential data assimilation methods, such as the Kalman filter (Law et al., 2015), allow us to recursively
update both the model state and the model parameters by conditioning them on observational data obtained
over sufficiently short time scales. Here, ensemble filtering methods provide a useful alternative to the
extended Kalman filter or variational methods; see Houtekamer and Zhang (2016) for a recent review of
various ensemble variants. With methods such as state augmentation, model parameters can be updated
as part of the filtering problem (Liu and West, 2001). Alternatively, the state values can be "integrated
out" to obtain the marginal likelihood over the model parameters; see Durbin and Koopman (2012). In
Hakkarainen et al. (2012), this so-called filter likelihood approach was used to estimate parameters of
chaotic systems. Yet essentially all filtering-based approaches introduce additional tuning parameters, such
as the length of the assimilation time window, the model error covariance matrix, and covariance inflation
parameters. These choices have an impact on model parameter estimation and may introduce bias. Indeed,
as discussed in Hakkarainen et al. (2013), changing the filtering method requires updating the parameters
of the dynamical model. Operational ensemble prediction systems (EPS) can be used even in the absence of
ensemble filtering. Parameter calibration methods such as those in Jarvinen et al. (2011); Laine et al. (2011)
have been applied to the Integrated Forecast System (IFS) weather models (Ollinaho et al., 2012, 2013,
2014) at European Centre for Medium-Range Weather Forecasts (ECMWF). However, these approaches
are heuristic and again limited to relatively short predictive windows.

Previous work (for example, Roeckner et al. (2003); ECMWF (2013); Stevens et al. (2013)) calibrates
parameters of climate models by matching summary statistics of quantities of interest, such as top-of-
atmosphere radiation, with the corresponding summary statistics from re-analysis data or output from
competing models. The vast majority of these approaches produce only point estimates. Järvinen et al.
(2010) infers the closure parameters of a large-scale computationally intensive climate model, ECHAM5,
using a Bayesian approach based on several summary statistics and realized via MCMC. However, com-
putational limitations make applying algorithms such as MCMC challenging. Even short MCMC chains of
roughly 2000 iterations require methods such as parallel sampling and early rejection for tractability (Solo-





nen et al., 2012). Moreover, even if these computational challenges can be overcome, finding statistics that actually constrain the parameters is difficult, and inference results can be thus be inconclusive. The failure of the summary statistic approach in Järvinen et al. (2010) can be explained intuitively: the chosen statistics average out too much information, and therefore fail to characterize the geometry of the underlying chaotic attractor in a meaningful way.

The present work combines two recent approaches that together enable a Bayesian approach to inference and uncertainty quantification for the parameters of chaotic high-dimensional models. The *correlation integral likelihood* (CIL) (Haario et al., 2015) is able to constrain the parameters of chaotic dynamical systems, and the *local approximation MCMC* (LA-MCMC) method for posterior sampling (Conrad et al., 2016) makes asymptotically exact posterior characterization feasible, even with computationally expensive models.

The CIL method is based on the concept of fractal dimension from mathematical physics, which, broadly speaking, characterizes the space-filling properties of the trajectory of a dynamical system. Earlier work (e.g., Cencini et al. (2010)) describes a number of different approaches for estimating the fractal dimension. Our previous work extends this concept: instead of computing the fractal dimension of a single trajectory, a similar computation measures the distance between different model trajectories (Haario et al., 2015). The modification provides a normally distributed summary statistic of the data, which is sensitive to changes in the underlying attractor from which the data was sampled. Statistics that better describe the attractor yield likelihood functions that can better constrain the model parameters and, therefore, more meaningful parameter posterior distributions.

A related approach, discussed in the context of intractable likelihoods, is Bayesian inference using synthetic likelihoods, proposed as an alternative to approximate Bayesian computation (ABC); see Wood (2010) and Price et al. (2018). The method we present here uses a different summary statistic. Moreover, the synthetic likelihood approach involves re-creating the likelihood for every new model parameter value, which would require excessive CPU times in our setting. Recent work by Morzfeld et al. (2018) describes another feature-vector approach for data assimilation. For more details and comparisons among these approaches, see the discussion below in Section 2.1.

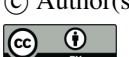
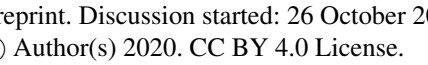


The LA-MCMC method (Conrad et al., 2016, 2018) approximates the computationally expensive log-likelihood function using local polynomial regression. In this method, the MCMC sampler directly uses the approximation of the log-likelihood to construct proposals and evaluate the Metropolis acceptance probability. Infrequently but regularly adding "full" likelihood evaluations to the point set used to construct the local polynomial regression continually improves the approximation, however. Expensive full likelihood evaluations are thus used only to construct the approximation or "surrogate" model. Conrad et al. (2016) show that, given an appropriate mechanism for triggering likelihood evaluations, the resulting Markov chain converges to the true posterior distribution while reducing the number of expensive likelihood evaluations (and hence forward model simulations) by orders of magnitude. Davis et al. (2020) show that LA-MCMC converges with approximately the expected $1/\sqrt{T}$ error decay rate after a finite number of steps $T$, and Davis (2018) introduces a numerical parameter that ensures convergence even if only noisy estimates of the target density are available. This modification is useful for the chaotic systems studied here.

The rest of this paper is organized as follows. Section 2 describes the methodologies used in this work, including the CIL, the stochastic LA-MCMC algorithm, and the merging of these two approaches. Section 3 is dedicated to numerical experiments. The approach is first verified by comparing results in cases where the parameter posteriors can also be obtained with standard sampling methods. Further modifications to reduce computational demand of evalutating the CIL are presented. A fine-grid version of the quasi-geostrophic (QG) model is then used to demonstrate that model identification is possible in long-range simulations, beyond reach of standard methods. These examples are followed by a concluding discussion in Section 4.

## 2 Methods

### 2.1 Correlation Integral Likelihood

We first construct a likelihood function that models the observations by comparing certain summary statistics of the observations to the corresponding statistics of a trajectory simulated from the chaotic model. As a source of statistics, we will choose the correlation integral, which depends on the fractal dimension of



the chaotic attractor. Unlike other statistics—such as the ergodic averages of a trajectory —the correlation

integral is able to constrain the parameters of a chaotic model (Haario et al., 2015).

Let us denote by

$$\frac{\mathrm{d}\mathbf{u}}{\mathrm{d}t} = f(\mathbf{u},\theta), \quad \mathbf{u}(t=0) = \mathbf{u}_0, \tag{1}$$

a dynamical system with state $\mathbf{u}(t) \in \mathbb{R}^n$, initial state $\mathbf{u}_0 \in \mathbb{R}^n$, and parameters $\theta \in \mathbb{R}^q$. The time-discretized system, with time steps $t_i \in \{t_1, \ldots, t_\tau\}$ denoting selected observation points, can be written as

$$\mathbf{u}_i \equiv \mathbf{u}(t_i) = F(t_i; \mathbf{u}_0, \theta). \tag{2}$$

Either the full state $\mathbf{u}_i \in \mathbb{R}^n$ or a subset $\mathbf{s}_i \in \mathbb{R}^{d \leq n}$ of the state components are observed. We will use $\mathbf{S} = \{\mathbf{s}_1, \ldots, \mathbf{s}_\tau\}$ to denote a collection of these observables at successive times.

Using the model–observation mismatch at a collection of times $t_i$ to constrain the value of the parameters $\theta$ is not suitable when the system (1) has chaotic dynamics, since the state vector values $\mathbf{s}_i$ are unpredictable

after a finite time interval. Though long-time trajectories $\mathbf{s}(t)$ of chaotic systems are not predictable in the time domain, they do, however, represent samples from an underlying attractor in the phase space. The states are generated deterministically, but the model's chaotic nature allows us to interpret the states as samples from a particular $\theta$-dependent distribution. Yet obvious choices for summary statistics $T$ that depend on the observed states $\mathbf{S}$, such as ergodic averages, ignore important aspects of the dynamics and

are thus unable to constrain the model parameters. For example, the statistic $T(\mathbf{S}) = \frac{1}{\tau}\sum_{i=1}^{\tau} \mathbf{s}_i$ is easy to compute and is normally distributed in the limit $\tau \to \infty$ (under appropriate conditions), but this ergodic mean says very little about the shape of the chaotic attractor.

Instead, we need a summary statistic that retains information relevant for parameter estimation, but still defines a computationally tractable likelihood. To this end, Haario et al. (2015) devised the correlation

integral likelihood (CIL), which retains enough information about the attractor to constrain the model parameters. We first review the CIL and then discuss how to make evaluation of the likelihood tractable.

We will use the CIL to evaluate the "difference" between two chaotic attractors. For this purpose, we will first describe how to statistically characterize the geometry of a *given* attractor, given suitable observations $\mathbf{S}$. In particular, constructing the CIL likelihood will require three steps: (i) computing distances between





observables sampled from a given attractor; (ii) evaluating the empirical cumulative distribution function (ECDF) of these distances and deriving certain summary statistics $T$ from the ECDF; and (iii) estimating the mean and covariance of $T$ by repeating steps (i) and (ii). Intuitively, the CIL thus interprets observations of a chaotic trajectory as samples from a fixed distribution over phase space. It allows the time between observations to be arbitrarily large—importantly, much longer than the system's non-chaotic prediction

window.

Now we describe the CIL construction in detail. Suppose that we have collected a data set $\mathbf{S}$ comprising observations of the dynamical system of interest. Let $\mathbf{S}$ be split into $n_{\mathrm{epo}}$ different subsets called *epochs*. The epochs can, in principle, be any subsets of length $N$ from the reference data set $\mathbf{S}$. In this paper, however, we restrict the epochs to be time-consecutive intervals of $N$ evenly spaced observations. Let

$\mathbf{s}^k = \{\mathbf{s}_i^k\}_{i=1}^N$ and $\mathbf{s}^l = \{\mathbf{s}_j^l\}_{j=1}^N$, with $1 \leq k, l \leq n_{\mathrm{epo}}$ and $k \neq l$, be two such disjoint epochs. The individual observable vectors $\mathbf{s}_i^k \in \mathbb{R}^d$ and $\mathbf{s}_j^l \in \mathbb{R}^d$ comprising each epoch come from time intervals $[t_{kN+1}, t_{(k+1)N}]$ and $[t_{lN+1}, t_{(l+1)N}]$, respectively. In other words, superscripts refer to different epochs and subscripts refer to the time points within those epochs. Haario et al. (2015) then define the *modified correlation integral sum* $C(R, N, \mathbf{s}^k, \mathbf{s}^l)$ by counting all pairs of observations that are less than a distance $R > 0$ from each

other:

$$C(R, N, \mathbf{s}^k, \mathbf{s}^l) = \frac{1}{N^2} \sum_{i,j \leq N} \mathbb{1}_{[0,R]}\left(\left\|\mathbf{s}_i^k - \mathbf{s}_j^l\right\|\right), \tag{3}$$

where $\mathbb{1}$ denotes the indicator function and $\|\cdot\|$ is the Euclidean norm on $\mathbb{R}^d$. In the physics literature, evaluating Eq. (3) in the limit $R \to 0$, with $k = l$ and $i \neq j$, numerically approximates the fractal dimension of the attractor that produced $\mathbf{s}^k = \mathbf{s}^l$ (Grassberger and Procaccia, 1983a, b). Here, we instead use Eq. (3)

to characterize the distribution of distances between $\mathbf{s}^k$ and $\mathbf{s}^l$ at all relevant scales. We assume that the state space is bounded; therefore, an $R_0$ covering all pairwise distances in Eq. (3) exists. For a prescribed set of radii $R_m = R_0 b^{-m}$, with $b > 1$ and $m = 0, \ldots, M$, Eq. (3) defines a discretization of the empirical CDF of the distances $\|\mathbf{s}_i^k - \mathbf{s}_j^l\|$, with discretization boundaries given by the numbers $R_m$.

Now we define $y_m^{k,l} = C(R_m, N, \mathbf{s}^k, \mathbf{s}^l)$ as components of a *statistic* $T(\mathbf{s}^k, \mathbf{s}^l) = \mathbf{y}^{k,l} := (y_0^{k,l}, \ldots, y_M^{k,l})$.

This statistic is also called the *feature vector*. According to Borovkova et al. (2001); Neumeyer (2004), the vectors $\mathbf{y}^{k,l}$ are normally distributed. This is a generalization of the classical result of Donsker (1951),

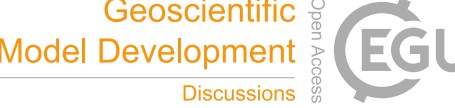

which applies to i.i.d. samples from a scalar-valued distribution. We characterize this normal distribution by subsampling the full data set $\mathbf{S}$. Specifically, we approximate the mean $\mu$ and covariance $\Sigma$ of $T$ by the sample mean and sample covariance of the set $\{\mathbf{y}^{k,l} : 1 \le k, l \le n_{\mathrm{epo}}, k \ne l\}$, evaluated for all $\frac{1}{2} n_{\mathrm{epo}}(n_{\mathrm{epo}}-1)$

pairs of epochs $(\mathbf{s}^k, \mathbf{s}^l)$ using fixed values of $R_0$, $b$, $M$, and $N$.

The Gaussian distribution of $T$ effectively characterizes the geometry of the attractor represented in the data set $\mathbf{S}$. Now we wish to use this distribution to infer the parameters $\theta$. Given a candidate parameter value $\tilde{\theta}$, we use the model to generate states $\mathbf{s}^*(\tilde{\theta}) = \{\mathbf{s}_i^*(\tilde{\theta})\}_{i=1}^N$ for the length of a single epoch. We then evaluate the statistics $y_m^{k,*} = C(R_m, N, \mathbf{s}^k, \mathbf{s}^*(\tilde{\theta}))$ as in Eq. (3), by computing the distances between

elements of $\mathbf{s}^*(\tilde{\theta})$ and the states of an epoch $\mathbf{s}^k$ selected from the data $\mathbf{S}$. Combining these statistics into a feature vector $\mathbf{y}^{k,*}(\tilde{\theta}) = (y_m^{k,*})_{m=0}^M$, we can write a noisy estimate of the log-likelihood function:

$$\log p(\tilde{\theta}|\mathbf{s}^k) = -\frac{1}{2}\left(\mathbf{y}^{k,*}(\tilde{\theta}) - \mu\right)^\top \Sigma^{-1} \left(\mathbf{y}^{k,*}(\tilde{\theta}) - \mu\right) + \text{constant} \tag{4}$$

Comparing $\mathbf{s}^*(\tilde{\theta})$ with other epochs drawn from the data set $\mathbf{S}$, however, will produce different realizations of the feature vector. We thus average the resulting log-likelihoods over *all* epochs,

$$\log p(\tilde{\theta}|\mathbf{S}) = \frac{1}{n_{\mathrm{epo}}} \sum_{k=1}^{n_{\mathrm{epo}}} \log p(\tilde{\theta}|\mathbf{s}^k). \tag{5}$$

This averaging, which involves evaluating Eq. (4) $n_{\mathrm{epo}}$ times, involves only new distance computations and is thus relatively cheap relative to time integration of the dynamical model.

Because the feature vectors $\mathbf{y}^{k,*}$ are random for any finite $N$, and because the number of epochs $n_{\mathrm{epo}}$ is also finite, the log-likelihood in Eq. (5) is necessarily random. It is then useful to view Eq. (5) as estimate

of an underlying *true* log-likelihood. We are therefore in a setting where cannot evaluate the unnormalized posterior density exactly; we only have access to a noisy approximation of it. Previous work (Springer et al., 2019) has demonstrated that derivative-free optimizers such as the differential evolution (DE) algorithm can successfully identify the posterior mode in this setting, yielding a point estimate of $\theta$. In the fully Bayesian setting, one could characterize the posterior $p(\theta|\mathbf{S})$ using pseudo-marginal MCMC methods (Andrieu and

Roberts, 2009), but at significant computational expense. Below, we will use a surrogate model constructed adaptively during MCMC sampling to reduce this computational burden.





Note that the CIL approach described above already reduces the computational cost of inference by only requiring simulation of the (potentially expensive) chaotic model for a single epoch. We compare *each* epoch of the data to the same single-epoch model output. Each of these comparisons results in an estimate of the log-likelihood, which we then average over data epochs. A larger data set $\mathbf{S}$ can reduce the variance of this average, but does not require additional simulations of the dynamical model. Also, we do not require any knowledge about the initial conditions of the model; we simply omit an initial time interval before extracting $\mathbf{s}^*(\tilde{\theta})$, to ensure that the observed trajectory is on the chaotic attractor.

Our approach is broadly similar to the synthetic likelihood method (e.g., Wood (2010); Price et al. (2018)), but differs in two key respects: (i) we use a novel summary statistic that is able to characterize chaotic attractors, and (ii) we only need to evaluate the forward model for a single epoch. Comparatively, synthetic likelihoods typically use summary statistics such as auto-covariances at a given lag or regression coefficients. These methods also require long-time integration of the forward model for each candidate parameter value $\theta$, rather than integration for only one epoch. Morzfeld et al. (2018) also discuss several ways of using feature vectors for inference in geophysics. A distinction of the present work is that we use an ECDF-based summary statistic that is provably Gaussian, and we perform extensive Bayesian analysis of the parameter posteriors via novel MCMC methods. These methods are described next.

## 2.2 Local Approximation MCMC

Even with the developments described above, estimating the correlation integral likelihood at each candidate parameter value $\tilde{\theta}$ can still be computationally intensive. We thus introduce a surrogate modeling method that replaces many of these CIL evaluations with an inexpensive approximation, while still providing convergence guarantees. This method is called local approximation MCMC (LA-MCMC).

First introduced in Conrad et al. (2016), LA-MCMC builds local surrogate models for the log-likelihood while *simultaneously* sampling the posterior. The surrogate is incrementally and infinitely refined during sampling and thus tailored to the problem—i.e., made more accurate in regions of high posterior probability. Specifically, the surrogate model is a local polynomial computed by fitting nearby evaluations of the "true" log-likelihood. Davis et al. (2020) shows that the error in the approximate Markov chain computed with the local surrogate model decays at approximately the expected $1/\sqrt{T}$ rate, where $T$ is the number of





MCMC steps. Davis (2018) demonstrated that noisy estimates of the likelihood are sufficient to construct
the surrogate model and still retain asymptotic convergence. Empirical studies (Conrad et al., 2016, 2018;
Davis et al., 2020) on problems of moderate parameter dimension showed that the number of expensive
likelihood evaluations per MCMC step can be reduced by orders of magnitude, with no discernable loss of
accuracy in posterior expectations.

Here we briefly summarize one step of the LA-MCMC construction, and refer to Davis et al. (2020) for
details. Each LA-MCMC step consists of four stages: (i) *possibly* refine the local polynomial approximation
of the log-likelihood, (ii) propose a new candidate MCMC state, (iii) compute the acceptance probability,
and (iv) accept or reject the proposed state. The major distinction between this algorithm and standard
Metropolis-Hastings MCMC is that the acceptance probability in stage (iii) is computed only using the
*approximation* or surrogate model of the log-likelihood, at both the current and proposed states. This intro-
duces an error, relative to computation of the acceptance probability with exact likelihood evaluations, but
stage (i) of the algorithm is designed to control and incrementally reduce this error at the appropriate rate.

"Refinement" in stage (i) consists of adding a computationally intensive log-likelihood evaluation at
some parameter value $\theta_i$, denoted by $\mathcal{L}(\theta_i)$, to the *evaluated set* $\{(\theta_i, \mathcal{L}(\theta_i))\}_{i=1}^K$. These $K$ pairs are used
to construct the local approximation via a kernel-weighted local polynomial regression (Kohler, 2002). The
values $\{\theta_i\}_{i=1}^K$ are called "support points" in this paper. Details on the regression formulation are in Davis
et al. (2020); Conrad et al. (2016). As the support points cover the regions of high posterior probability
more densely, the accuracy of the local polynomial surrogate will increase. This error is well understood
(Kohler, 2002; Conn et al., 2009) and, crucially, takes advantage of smoothness in the underlying true
log-likelihood function. This smoothness ultimately allows the cardinality of the evaluated set to be much
smaller than the number of MCMC steps.

Intuitively, if the surrogate converges to the true log-likelihood, then the samples generated with LA-
MCMC will (asymptotically) be drawn from the true posterior distribution. After any finite number of
steps, however, the surrogate error introduces a bias into the sampling algorithm. The refinement strategy
must therefore ensure that this bias is not the dominant source of error. At the same time, refinements
must occur infrequently, to ensure that LA-MCMC is computationally cheaper than using the true log-
likelihood. Davis et al. (2020) analyzes the trade-off between surrogate-induced bias and MCMC variance





and proposes a rate-optimal refinement strategy. We use essentially the same algorithm here, but add an isotropic $\ell^2$ penalty on the polynomial coefficients of the kernel-weighted local regression problem solved to evaluate the surrogate at any parameter value $\theta$; in other words, we perform local *ridge regression* rather than ordinary least squares, which improves performance with noisy likelihoods.

For pseudocode of the full LA-MCMC algorithm, we refer to Davis et al. (2020, Algorithm 1). Parameters of the algorithm are as follows: (i) initial error threshold $\gamma_0 = 1$; (ii) error threshold decay rate $\gamma_1 = 1.0$; (iii) maximum poisedness constant $\bar{\Lambda} = 100$; (iv) tail-correction parameter $\eta = 0$ (no tail correction); (v) local polynomial degree $p = 2$. The number of nearest neighbors $k$ used to construct each local polynomial surrogate is chosen to be $k = k_0 + (K - k_0)^{1/3}$ where $k_0 = \sqrt{q}D$, $q$ is the dimension of the parameters $\theta \in \mathbb{R}^q$, and $D$ is the number of coefficients in the local polynomial approximation of total degree $p = 2$, i.e., $D = (q+2)(q+1)/2$. If we had $k = D$, the approximation would be an interpolant. Instead we oversample by a factor $\sqrt{q}$, as suggested in Conrad et al. (2016), and allow $k$ to grow slowly with the size $K$ of the evaluated set as in Davis (2018).

## 3 Numerical experiments

This section contains numerical experiments to illustrate the methods introduced in the previous sections. As a large-scale example, we characterize the posterior distribution of parameters in the two-layer quasi-geostrophic (QG) model. The computations needed to characterize the posterior distribution with standard MCMC methods in this example would be prohibitive without massive computational resources and are therefore omitted. In contrast, we will show that the LA-MCMC method is able to simulate from the parameter posterior distribution.

Before presenting this example, we first demonstrate that the posteriors produced by LA-MCMC agree with those obtained via exact MCMC sampling methods in cases where the latter are computationally tractable, using two examples: the classical Lorenz-63 system and the higher-dimensional Kuramoto–Shivashinsky (KS) model. In both of these examples, we quantify the computational savings due to LA-MCMC, and in the second we introduce additional ways to enhance computation using parallel (GPU) integration.




Let $\Delta_t$ denote the time difference between consecutive observations; one epoch thus contains the times in the interval $[iN\Delta_t, (i+1)N\Delta_t)$. The number of data points in one epoch $N$ varies between 1000 and

2000, depending on the example. The training set $\mathbf{S}$ consists of a collection of $n_{\mathrm{epo}}$ such intervals. For numerical tests one can either use one long time series or integrate a shorter time interval several times using different initial values to create the training set for the likelihood. For these experiments the latter method was used with $n_{\mathrm{epo}} = 64$, yielding $\binom{n_{\mathrm{epo}}}{2} = 2016$ different pairs $(\mathbf{s}^k, \mathbf{s}^l)$, each of which resulted in an ECDF constructed from $N^2$ pairwise distances. According to tests performed while calibrating the

algorithm, these values of $N$ and $n_{\mathrm{epo}}$ are sufficient to obtain robust posterior estimates. With less data, the parameter posteriors will naturally be less precise.

The range of the bin radii $R_m$, $m = 0, \ldots, M$ is selected by examining the distances within the training set, keeping in mind that a positive variance is needed for every bin to avoid a singular covariance matrix. So the largest radius $R_0$ can be obtained from

$$R_0 = \min_{k \neq l} \left\{ \max_{i,j} \left\| s_i^k - s_j^l \right\| \right\} \tag{6}$$

over the disjoint subsets of the samples $\mathbf{s}^k$ and $\mathbf{s}^l$ of length $N$. As always with histograms, the number of bins $M$ must be selected first. The smallest radius is selected by requiring that for all the possible pairs $(\mathbf{s}^k, \mathbf{s}^l)$, it holds that $\mathcal{B}_{R_M}(\mathbf{s}_i^k) \cap \mathbf{s}^l \neq \emptyset$, where $\mathcal{B}_{R_M}(\mathbf{s}_i^k)$ is the ball of radius $R_M$ centered at $\mathbf{s}_i^k$. That is,

$$R_M = \max_{k \neq l} \left\{ \min_{i,j} \left\| s_i^k - s_j^l \right\| \right\} \tag{7}$$

The base value $b$ is obtained by $R_M = R_0 b^{-M}$ and via this value, we fix all the other radii $R_m$. In the examples presented in this section, the length $M$ of the feature vector is 14 for the Lorenz-63 model and 32 for the higher-dimensional KS and QG models.

To balance the possibly different magnitudes of the components of the state vector, each component is scaled and shifted to the range $[-1, 1]$ before computing the distances. While this scaling could also be

performed in other ways, this method worked well in practice for the models considered. The normality of the ensemble of feature vectors is ascertained by comparing the histograms of the quadratic forms in Eq. (4) visually to the appropriate $\chi^2$ distribution.





In all the three experiments we create MCMC chains of length $10^5$. However, due to the use of the LA-MCMC approach, the number of full forward model evaluations is much lower, around 1000 or less; we

will report these values more specifically below.

### 3.1 Lorenz 63

We use the classical three-dimensional Lorenz 63 system (Lorenz, 1963) as a simple first example to demonstrate how LA-MCMC can be successfully paired with the CIL and the Adaptive Metropolis (AM) algorithm (Haario et al., 2001, 2006) to obtain the posterior distribution for chaotic systems at a greatly

reduced computational cost, compared to AM without the local approximation. The time evolution of the state vector $\mathbf{s} = (X, Y, Z)$ is given by

$$
\begin{aligned}
\dot{X} &= \sigma(Y - X), \\
\dot{Y} &= X(\rho - Z) - Y, \\
\dot{Z} &= XY - \beta Z.
\end{aligned}
\tag{8}
$$

This system of equations is often said to describe an extreme simplification of a weather model.

The reference data were generated with parameter values $\sigma = 10$, $\rho = 28$, and $\beta = \frac{8}{3}$ by performing $n_{\text{epo}} = 64$ distinct model simulations, with observations made at 2000 evenly distributed times between $[10, 20000]$. These observations were perturbed with $5\%$ multiplicative Gaussian noise. The length of the predictable time window is roughly 7, which is less than the time between consecutive observations. The

parameters of the CIL method were obtained as described at the start of Section 3, with values $M = 14$, $R_0 = 2.85$, and $b = 1.51$.

The set of vectors $\{\mathbf{y}^{k,l} | k, l \leq n_{\text{epo}}\}$ is shown in Fig. 1 in the log-log scale. The figure shows how the variability of these vectors is quite small. Figure 1 validates the normality assumption for feature vectors.

Pairwise two-dimensional marginals of the parameter posterior are shown in Fig. 2, both from sampling

the posterior with the full forward model and with using the surrogate model approach for generating the chain. These two posteriors are almost perfectly superimposed.

To get an idea of the computational savings achieved with LA-MCMC, the computation of the MCMC chains of length $10^5$ was repeated 10 times. The cumulative number of full likelihood evaluations is pre-




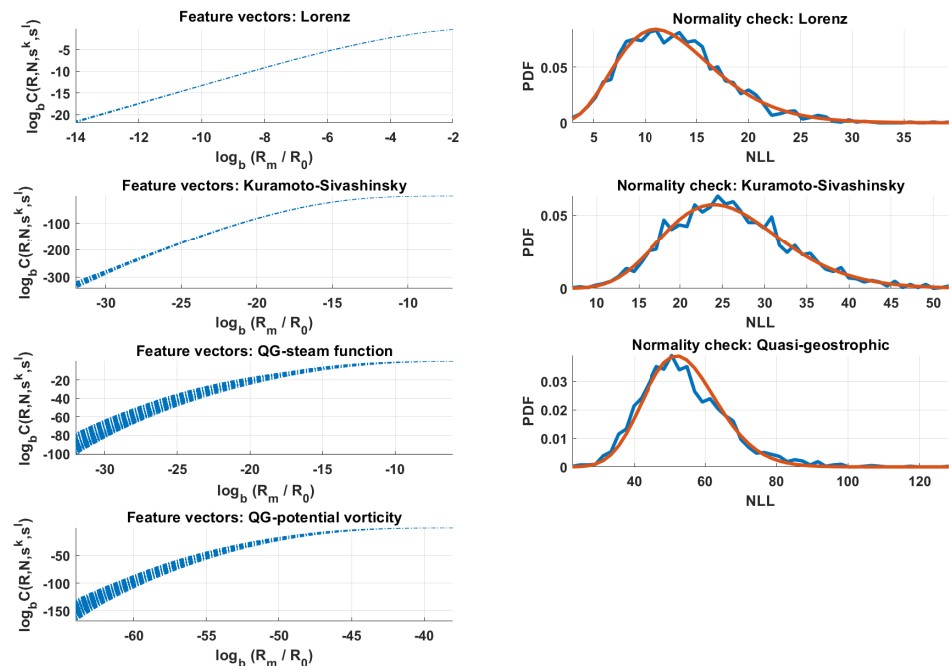

**Figure 1.** Left. For all combinations of $k$ and $l$, the feature vectors for the Lorenz-63, the Kuramoto-Sivashinsky and concatenated feature vectors for the quasi-geostrophic system are shown. Right. Normality check: the $\chi^2$ density function versus the histograms of the respective negative log-likelihood (NLL) values; see Eq. (4).

sented in Fig. 3. At the end of the chains, the number of full likelihood evaluations varied between 955 and
1016. Thus by using LA-MCMC in this setting, remarkable computational savings of up to two orders of magnitude are achieved.



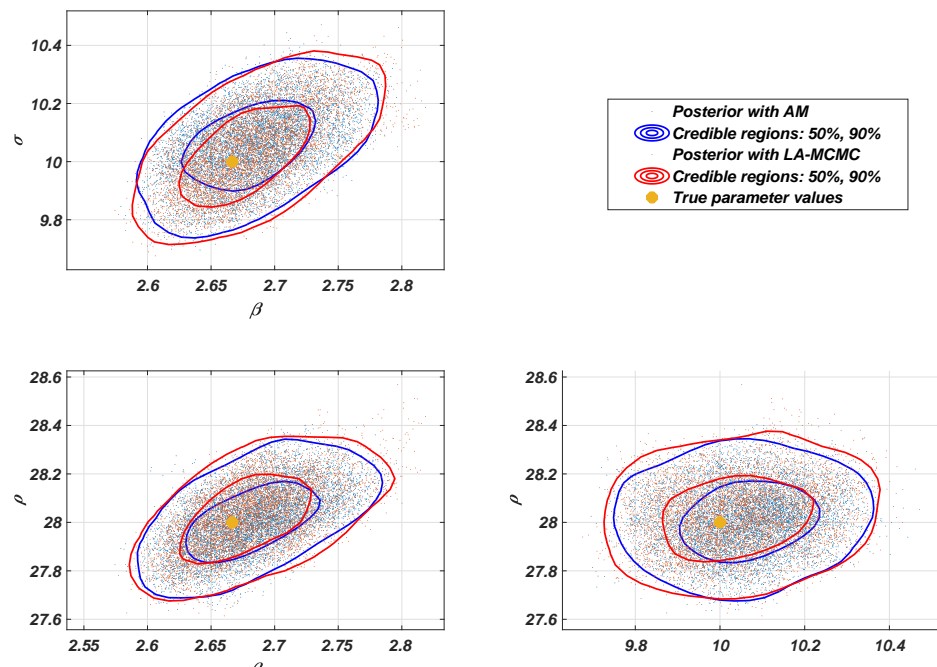

**Figure 2.** Two-dimensional posterior marginal distributions of the parameters of the Lorenz 63 model obtained with LA-MCMC and AM.

### 3.2 The Kuramoto–Sivashinsky model

The second example is the 256-dimensional Kuramoto–Sivashinsky (KS) PDE system. The purpose of this example is to introduce ways to improve the computational efficiency by a piecewise parallel integra-
tion over the time interval of given data. Also, we demonstrate how decreasing the number of observed components impacts the accuracy of parameter estimation. Even though the posterior evaluation proves to be relatively expensive, direct verification of the results with those obtained by using standard adaptive



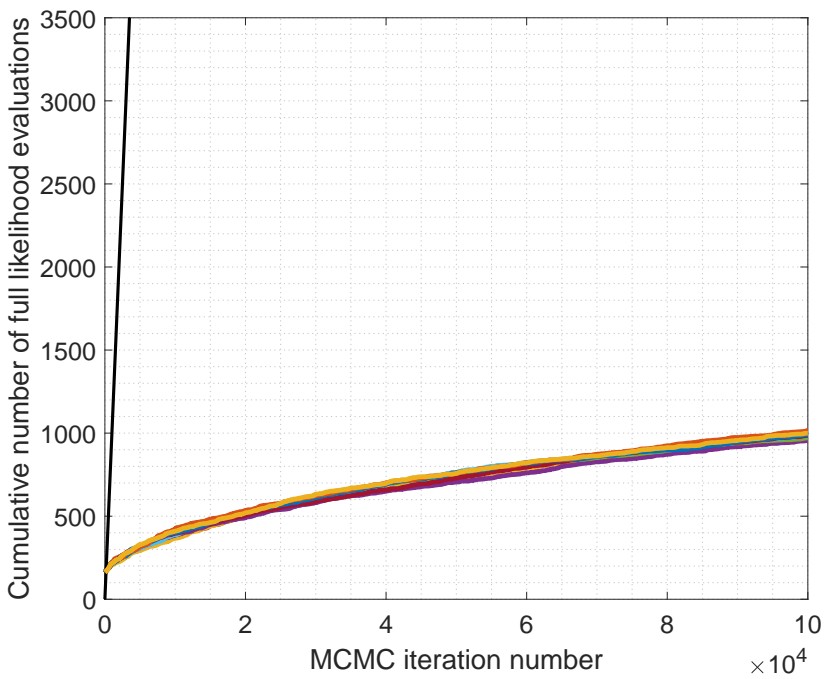

**Figure 3.** Comparison of the cumulative number of full likelihood evaluations while using AM (black line) and LA-MCMC (colored lines). Every colored line correspond to a different chain obtained with LA-MCMC by using the same likelihood.

MCMC is still possible. The Kuramoto-Sivashinsky model is given by the fourth-order PDE

$$s_t = -ss_x - \frac{1}{\eta}s_{xx} - \gamma s_{xxxx}, \tag{9}$$

where $s = s(x,t)$ is a real function of $x \in \mathbb{R}$ and $t \in \mathbb{R}_+$. In addition it is assumed that $s$ is spatially periodical with period of $L$, i.e. $s(x+L,t) = s(x,t)$. This experiment uses the parametrization from (Yiorgos Smyrlis, 1996) that maps the spatial domain $\left[-\frac{L}{2}, \frac{L}{2}\right]$ to $[-\pi, \pi]$ by setting $\tilde{x} = \frac{2\pi}{L}x$ and $\tilde{t} = \left(\frac{2\pi}{L}\right)^2 t$. With $L = 100$, the true value of parameter $\gamma$ is $(\pi/50)^2 \approx 0.0039$, and the true value of $\eta$ becomes $\frac{1}{2}$. These two parameters are the ones that are then estimated with the LA-MCMC method. This system was





derived by Kuramoto et al. in (Kuramoto and Yamada, 1976; Kuramoto, 1978) as a model for phase turbu-
lence in reaction-diffusion systems. (Sivashinsky, 1977) used the same system for describing instabilities
of laminar flames.

Assume that the solution for this problem can be represented by a truncation of the Fourier series

$$s(x,t) = \sum_{j=0}^{\infty} \left[ A_j(t) \sin\left(\frac{2\pi}{L}jx\right) + B_j(t) \cos\left(\frac{2\pi}{L}jx\right) \right]. \tag{10}$$

Using this form reduces Eq. (9) to a system of ordinary differential equations for the unknown coefficients
$A_j(t)$ and $B_j(t)$,

$$\dot{A}_j(t) = \alpha_1 j^2 A_j(t) + \alpha_2 j^4 A_j(t) + F_1(A(t)) \tag{11}$$

$$\dot{B}_j(t) = \beta_1 j^2 B_j(t) + \beta_2 j^4 B_j(t) + F_2(B(t)), \tag{12}$$

where the terms $F_1(\cdot)$ and $F_2(\cdot)$ are polynomials of the vectors $A$ and $B$. For details, see (Huttunen et al.,
2018). The solution can be effectively computed on graphics processors in parallel, and if computational
resources allow, several instances of Eq. (9) can be solved in parallel. Even on fast consumer-level lap-
tops, several thousand simulations can be performed in parallel when the discretization of the $x$-dimension
contains around 500 points.

A total of 64 epochs of the 256-dimensional KS model are integrated over the time interval $[0, 150000]$,
and as in the case of the Lorenz-63 model, the initial predictable time window is discarded and $1024$
equidistant measurements from $[500, 150000]$ are selected, with $\Delta_t \approx 146$. The parameters used for the
CIL method were $R_0 = 1801.7$, $M = 32$, and $b = 1.025$.

The time needed to integrate the model up to $t = 150000$ is approximately 103 seconds with the Nvidia
1070 GPU, implying that generating an MCMC chain with 100000 samples with standard MCMC algo-
rithms would take almost four months. The use of LA-MCMC alone again shrinks the time needed by a
factor of 100, that is, to around 28 hours. However, the calculations can yet be considerably enhanced by
parallel computing. In practice this translates the problem of generating a candidate trajectory of length
150000 into generating observations from several shorter time intervals. In our example, an efficient divi-
sion is to perform 128 parallel calculations each of length 4500, with randomized initial values close to the
values selected from the training set. Discarding the predictable interval $[0, 500]$ and taking 8 observations





**Table 1.** Parameter values of the four parameter vectors used in the forward KS model simulation examples in Fig. 5. The parameter vector in the first column labeled 1 are the true parameters, and the second one resides inside the posterior. The last two are outside the posterior. These parameters correspond to points shown in the posterior distribution shown in Fig. 4.

|  | Case 1 | Case 2 | Case 3 | Case 4 |
|---|---|---|---|---|
| $\eta$ | 0.50000 | 0.47820 | 0.49500 | 0.52000 |
| $\gamma$ | 0.00395 | 0.00467 | 0.00350 | 0.00500 |

at intervals of 500 yields the same number 1024 of observations as in the initial setting. While the total integration time increases, this reduces the wall clock time needed for computation of a single candidate simulation from 103 s to 2.5 s. The full MCMC chain can be then be generated in 70 hours without the surrogate model, and in 42 minutes using LA-MCMC.

Parameter posterior distributions from the KS system, produced with MCMC both with and without the local approximation surrogate, are shown in Fig. 4. Repeating the calculations several times yielded no meaningful differences in the results. In this experiment, the number of forward model evaluations LA-MCMC needed for generating a chain of length 100000 was in the range $[1131, 1221]$.

Model trajectories from simulations with four different parameter vectors are shown in Fig. 5. These
parameter values were 1) the "true" value which was used to generate training data, 2) another parameter from inside the posterior distribution, and 3-4) two other parameters from outside the posterior distribution. These parameters are also shown in Fig. 4. Visually inspecting the outputs, Cases 1–3 look similar, while results using parameter vector 4, furthest away from the posterior, are markedly different. Even though the third parameter vector is outside the posterior, the resulting trajectory is not easily distinguishable from 1
and 2, indicating that the CIL method differentiates between the trajectories more efficiently.

Additional experiments were performed to evaluate the stability of the method when not all of the model states were observed. Keeping the setup otherwise fixed, the number of elements of the state vectors observed was reduced from the full 256 step by step to 128, 64, and 32. The resulting MCMC chains are presented in Fig. 6, and as expected, when less is observed, the size of the posterior distribution grows.





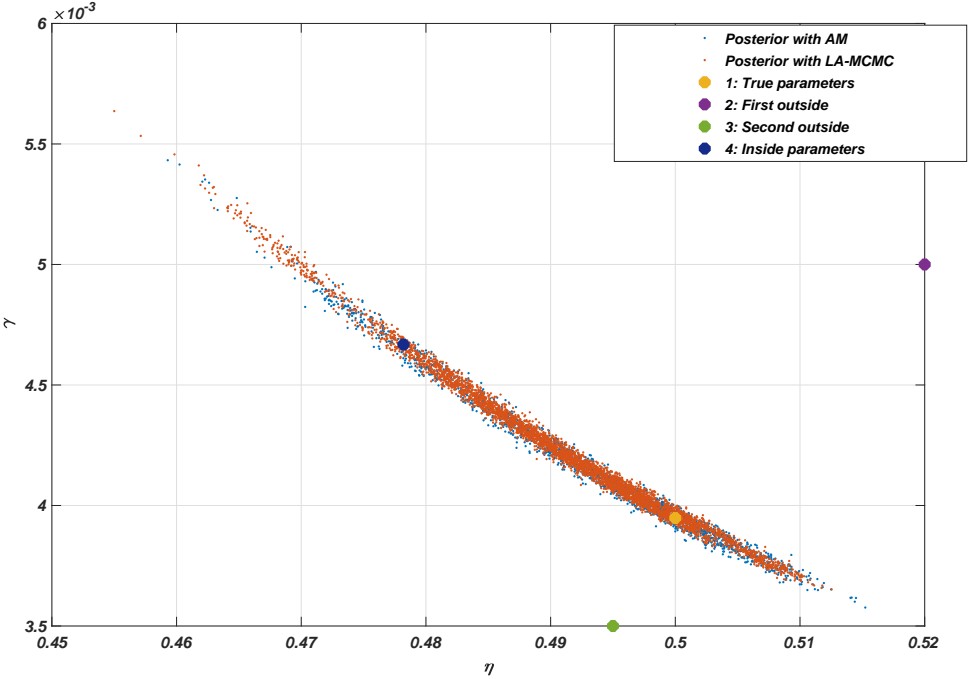

**Figure 4.** Posterior distribution of the parameters of the KS system. The parameter values are shown in Table 1, while examples of the respective integrated trajectories are given in Fig. 5.

### 3.3 The quasi-geostrophic model

The methodology is here applied to a computationally intensive model, where a brute-force parameter posterior estimation would be too time-consuming. We employ the well-known quasi-geostrophic model ((Fandry and Leslie, 1984) and (Pedlosky, 1987)) using a dense grid to achieve complex chaotic dynamics in high dimensions. The wall-clock time for one forward model simulation is roughly 10 minutes, so a naïve calculation of a posterior sample of size 100000 would take around two years. We demonstrate how the application of the methods verified in the two previous examples reduces this time to a few hours.



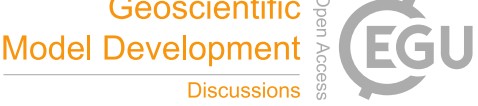

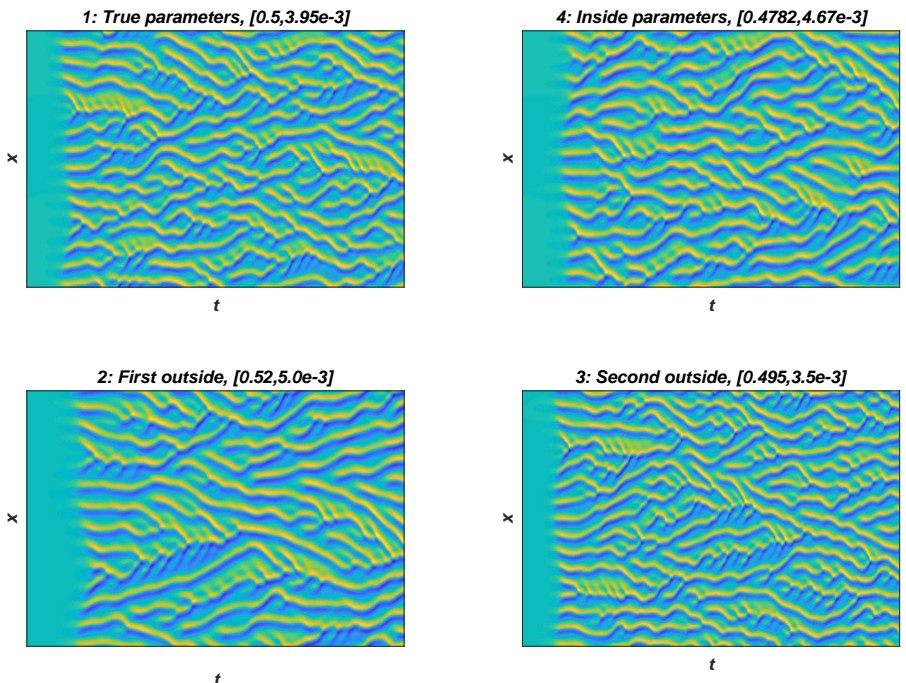

**Figure 5.** Example model trajectories from the KS system. Figure 1) shows simulation using the true parameters, the parameters used for figure 4) are inside the posterior distribution, and figures 2) and 3) are generated from simulations with parameters outside the posterior distribution, shown in Fig. 4. The values of the parameter vectors 1, 2, 3 and 4 are given in Table 1. The $y$-axis shows the 256-dimensional state vector, and the $x$-axis the time evolution of the system.

The two-layer quasi-geostrophic model simulates winds on a cylindrical surface vertically divided into two interacting layers, where the bottom layer is influenced by a prescribed topography. The model geometry implies periodic boundary conditions, seamlessly stitching together the extreme eastern and western parts of the rectangular spatial domain with coordinates $x$ and $y$. For the northern and southern edges, user-specified time-independent Dirichlet boundary conditions are used. In addition to these conditions


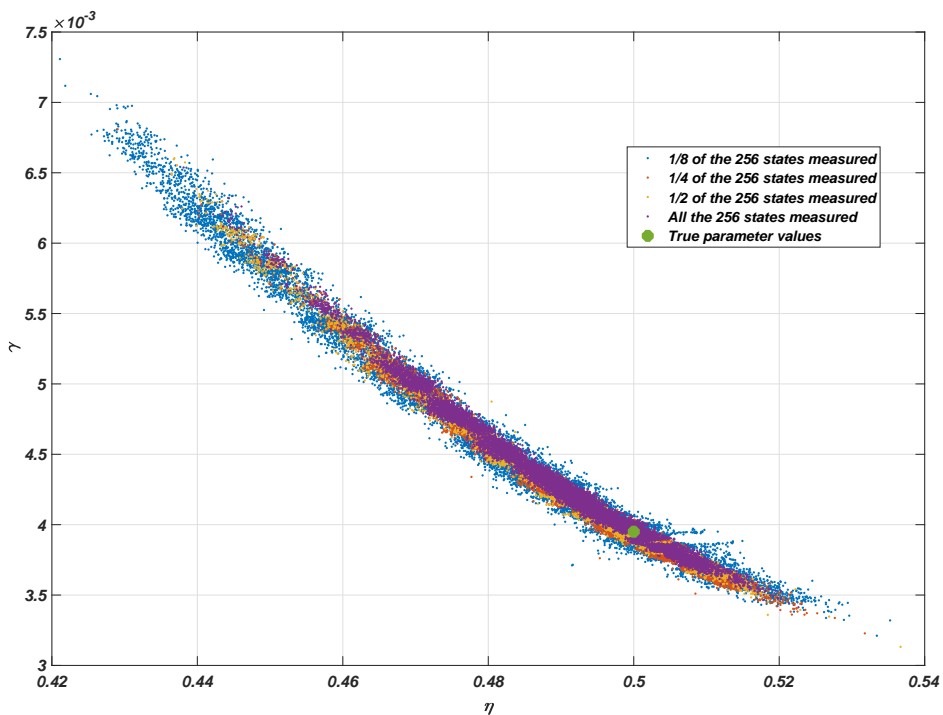

**Figure 6.** Comparison between the KS system's posterior distribution in cases where all or only a part of the states are observed.

and the topographic constraints, the model parameters include the thicknesses of the two atmospheric layers, denoted by $H_1$ and $H_2$. Furthermore, the QG-model accounts for the Coriolis force, whose strength is controlled by the parameter $f_0$. An example of the two-layer geometry is presented in Fig. 7.

In a non-dimensional form the QG system can be written as

$$q_1 = \Delta\psi_1 - F_1(\psi_1 - \psi_2) + \beta y, \tag{13}$$

$$q_2 = \Delta\psi_2 - F_2(\psi_2 - \psi_1) + \beta y + R_s, \tag{14}$$




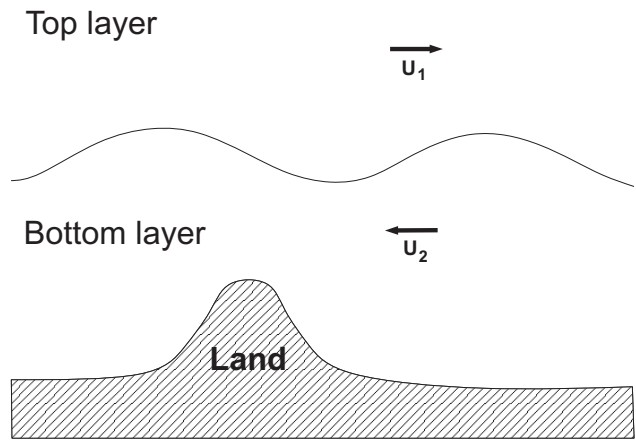

**Figure 7.** An example of the layer structure of the two-layer quasi-geostrophic model. The terms $U_1$ and $U_2$ denote mean zonal flows respectively in the top and the bottom layer.

where $q_i$ are potential vorticities, and $\psi_i$ are stream functions with indexes $i = 1, 2$ for the upper and the lower layers, respectively. Both the $q_i$ and $\psi_i$ are functions of time $t$ and spatial coordinates $x$, and $y$.

The coefficients $F_i = \frac{f_0^2 L^2}{\acute{g} H_i}$ control how much the model layers interact, $\beta$ is the northward gradient of the Coriolis force that gives rise to faster cyclonic flows closer to the poles, $L$ is a length scale constant and $\acute{g}$ is a gravity constant. Finally, $R_s(x, y) = \frac{S(x,y)}{\eta H_2}$ defines the topography for the lower layer where $\eta = \frac{U}{f_0 L}$ is the Rossby number of the system with $L$ and $U$ designating the length and speed scales, respectively. For further details, see (Fandry and Leslie, 1984) and (Pedlosky, 1987).

It is assumed that the motion determined by the model is geostrophic, essentially meaning that potential vorticity of the flow is preserved on both layers:

$$\frac{\partial q_i}{\partial t} + u_i \frac{\partial q_i}{\partial x} + v_i \frac{\partial q_i}{\partial y} = 0. \tag{15}$$

Here $u_i$ and $v_i$ are velocity fields, which are functions of both space and time. They are obtained from the stream functions $\psi_i$ via

$u_i = -\frac{\partial \psi_i}{\partial y}, \qquad v_i = \frac{\partial \psi_i}{\partial x}. \tag{16}$

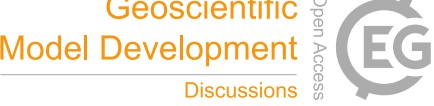

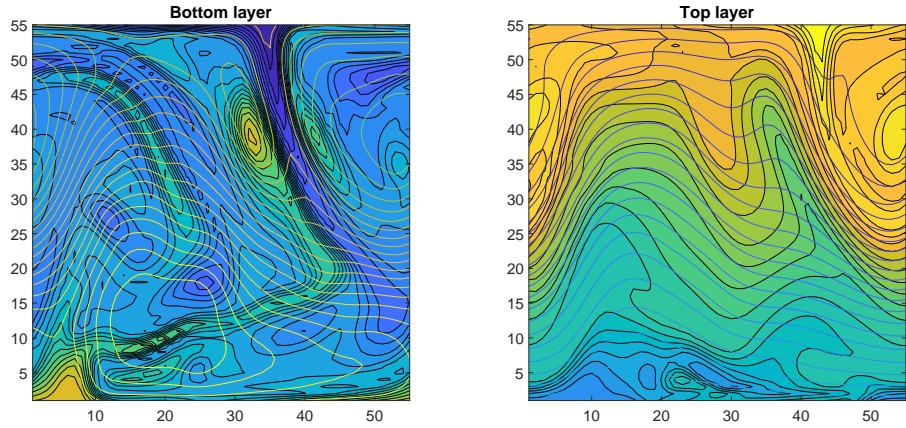

**Figure 8.** An example of the 6050-dimensional state of the quasi-geostrophic model. The contour lines for both the stream function and potential vorticity are shown for both layers. Note the cylindrical boundary conditions.

Equations (13)–(16) define the spatio-temporal evolution of the quantities $q_i, \psi_i, i = 1, 2$.

The numerical integration of this system is carried out using the semi-Lagrangian scheme, where the potential vorticities $q_i$ are computed according to Eq. (15) for given velocities $u_i$ and $v_i$. With these $q_i$ the stream functions can then be obtained from Eq. (13) and (14) with a two-stage finite difference scheme.

Finally, the velocity field is updated by Eq. (16) for the next iteration round. For more details see (Fandry and Leslie, 1984).

For estimating model parameters from synthetic data, a reference data set is created with 64 epochs each containing $N = 1000$ observations. These data are sampled from the model trajectory with $\Delta_t = 8$ (where a time step of length 1 corresponds to 6h) in the time interval $[192, 8192]$, that amounts to a long-range

integration of roughly 5–6 years of a climate model. The spatial domain is discretized into a $55 \times 55$ grid, which results in consistent chaotic behavior and more complex dynamics than with the often-used $20 \times 40$ grid. This is reflected in higher variability in the feature vectors, as seen in the Fig. 1. A snapshot of the 6050-dimensional trajectory of the QG system is displayed in Figure 8.



The model state is characterized by two distinct fields, the vorticities and stream functions, that naturally
are dependent on each other. But as shown in (Haario et al., 2015), it is useful to construct separate feature
vectors to characterize the dynamics in such situations. For this reason, two separate feature vectors are
constructed – one for the potential vorticity on both layers and the other for the stream function.

The Gaussian likelihood of the state is created by stacking these two feature vectors one after another.
The normality of the resulting $2(M+1)$-dimensional vector may again be verified as shown in Fig. 1. The
number of bins was set to 32, leading to parameter values $R_0 = 55$ and $b = 1,075$ for potential vorticity,
and $R_0 = 31$, and $b = 1.046$ for the stream function.

For parameter estimation, inferring the layer heights from synthetic data is considered. The reference
data set with $n_{\text{epo}} = 64$ integrations is produced using the values $H_1 = 5500$ and $H_2 = 4500$. A single for-
ward model evaluation takes 10 minutes on a fast laptop. So brute force MCMC chains of length 100000
would take around 2 years to run. But again the use of LA-MCMC reduces the computational time with a
factor of 100. In the experiments performed, the number of forward model evaluations needed was rang-
ing in the interval $[682, 762]$, which translates to around one week of computing time. As verified with
Kuramoto-Sivashinsky example, the forward model integration can be split to segments computed in par-
allel, which reduced time required to generate data for computing the likelihood further with a factor around
50, corresponding to around 3 h for generating the MCMC chain. The pairwise distances for generating
the feature vectors were computed on a GPU and therefore the required computation time for doing this
was negligible compared to the model integration time. The posterior distribution of the two parameters is
presented in Fig. 9.

## 4 Conclusions and future work

Bayesian parameter estimation with computationally demanding computer models is highly non-trivial.
The associated computational challenges often become insurmountable when the model dynamics are
chaotic. In this work we showed it is possible to overcome these challenges by combining the correlation
integral likelihood (CIL) with an MCMC method based on local surrogates of the log-likelihood function
(LA-MCMC). The CIL captures changes in the geometry of the underlying attractor of the chaotic system,



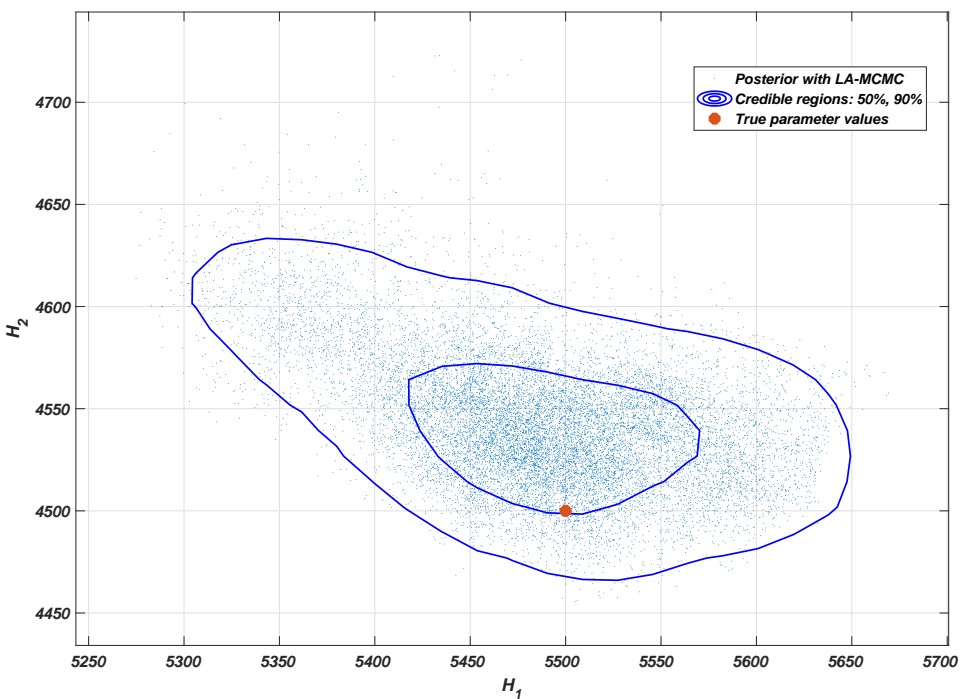

**Figure 9.** The clearly non-Gaussian posterior distribution of the $H_1$ and $H_2$-parameters of the quasi-geostrophic system shows how these parameters anticorrelate with each other.

460   while local approximation MCMC makes generating long MCMC chains based on this likelihood tractable, with computational savings of roughly two orders of magnitude in the experiments shown. Our methods were verified by sampling the parameter posteriors of the Lorenz-63 and the Kuramoto–Sivashinsky models, where an (expensive) comparison to exact MCMC with the CIL was still feasible. Then we applied our approach to the quasi-geostrophic model with a deliberately extended grid size. Without CIL, param-

465   eter estimation would not have been possible with chaotic models such as these; without LA-MCMC, the generation of long MCMC and sufficiently accurate chains for the higher-resolution QG model parameters





would have been computationally intractable. We note that the computational demands of the QG model already get quite close to those of weather models at coarse resolutions.

There are many potential directions for extension of this work. First, it should be feasible to run *parallel* LA-MCMC chains that share model evaluations in a single evaluated set; doing so can accelerate the construction of accurate local surrogate models, as demonstrated in Conrad et al. (2018), and is a useful way of harnessing parallel computational resources within surrogate-based MCMC. Extending this approach to higher-dimensional parameters is also of interest. While LA-MCMC has been successfully applied to chains of dimension up to $q = 12$ (Conrad et al., 2018), future work should explore sparsity and other truncations of the local polynomial approximation to improve scaling with dimension. From the CIL perspective, calibrating more complex models, such as weather models, often requires choosing the part of the state vector from which the feature vectors are computed. While computing the likelihood from the full high-dimensional state is computationally feasible, Haario et al. (2015) showed that carefully choosing a subset of the state for the feature vectors performs better. Also, the epochs may need to be chosen sufficiently long to include potential rare events, so that changes in rare event patterns can be identified. This, naturally, will increase the computational cost if one wants to be confident of the inclusion of such events.

While answering these questions will require further work, we believe the research presented in this paper provides a promising and reasonable step towards estimating parameters in the context of expensive operational models.

*Code availability.* Code is available at request.

*Author contributions.* HH and YM designed the study with input from all authors. SS, HH, AD, and JS combined the CIL and LA-MCMC methods for carrying out the research. AB wrote and provided implementations of the KS and QG models for GPUs, including custom numerics and testing. SS wrote the CIL code and the version of LA-MCMC used (based on earlier work by Antti Solonen), and carried out the simulations. All authors discussed the results and shared the responsibility of writing the manuscript. SS prepared the figures.





*Competing interests.* The authors declare that they have no conflict of interest

*Acknowledgements.* This work was supported by the Centre of Excellence of Inverse Modelling and Imaging (CoE), Academy of Finland, decision number 312 122. YM and AD were supported by the US Department of Energy, Office of Advanced Scientific Computing Research (ASCR), SciDAC program, as part of the FASTMath Institute.

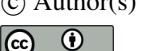



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
