# Peer review of "Efficient Bayesian inference for large chaotic dynamical systems"

_Geoscientific Model Development, 2020_

## Short Comment (SC1) · 14 Nov 2020

Dear authors,

in my role as Executive editor of GMD, I would like to bring to your attention our Editorial version 1.2:

https://www.geosci-model-dev.net/12/2215/2019/

This highlights some requirements of papers published in GMD, which is also available on the GMD website in the 'Manuscript Types' section: http://www.geoscientific-model-development.net/submission/manuscript_types.html

In particular, please note that for your paper, the following requirement has not been

met in the Discussions paper:

- "The main paper must give the model name and version number (or other unique identifier) in the title."

- Code must be published on a persistent public archive with a unique identifier for the exact model version described in the paper or uploaded to the supplement, unless this is impossible for reasons beyond the control of authors. All papers must include a section, at the end of the paper, entitled "Code availability". Here, either instructions for obtaining the code, or the reasons why the code is not available should be clearly stated. It is preferred for the code to be uploaded as a supplement or to be made available at a data repository with an associated DOI (digital object identifier) for the exact model version described in the paper. Alternatively, for established models, there may be an existing means of accessing the code through a particular system. In this case, there must exist a means of permanently accessing the precise model version described in the paper. In some cases, authors may prefer to put models on their own website, or to act as a point of contact for obtaining the code. Given the impermanence of websites and email addresses, this is not encouraged, and authors should consider improving the availability with a more permanent arrangement. Making code available through personal websites or via email contact to the authors is not sufficient. After the paper is accepted the model archive should be updated to include a link to the GMD paper.

This open source – open code strategy is also valid for articles of the paper type "methods for assessment of models". Therefore a code availability section stating only "Code is available at request." is not sufficient for a publication of your article, you have to provide access to the code unless license issue prevent this. In the latter case you have to clearly state which license issues.

Yours, Astrid Kerkweg

---

## Author Comment (AC1) · 7 Dec 2020

- "The main paper must give the model name and version number (or other uniquei-dentifier) in the title." - Code must be published on a persistent public archive with a unique identifier forthe exact model version described in the paper or uploaded to the supplement,unless this is impossible for reasons beyond the control of authors. All papersmust include a section, at the end of the paper, entitled "Code availabil-ity". Here,either instructions for obtaining the code, or the reasons why the code is notavailable should be clearly stated. It is preferred for the code to be uploaded asa supplement or to be made available at a data repository with an associatedDOI (digital object identifier) for the exact model version described in the paper.Alternatively, for established models, there may be an existing means of access-ing the code through

a particular system. In this case, there must exist a meansof permanently accessing the precise model version described in the paper. Insome cases, authors may prefer to put models on their own website, or to act as apoint of contact for obtaining the code. Given the impermanence of websites andemail addresses, this is not encouraged, and authors should consider improv-ing the availability with a more permanent arrangement. Making code availablethrough personal websites or via email contact to the authors is not sufficient.After the paper is accepted the model archive should be updated to include a linkto the GMD paper. This open source – open code strategy is also valid for articles of the paper type "meth-ods for assessment of models". Therefore a code availability section stating only "Codeis available at request." is not sufficient for a publication of your article, you have to pro-vide access to the code unless license issue prevent this. In the latter case you haveto clearly state which license issues. Yours, Astrid Kerkweg ——————————————————————————————————————
- Dear Astrid Kerkweg,

We certainly will provide the code available for the readers. We would prefer to upload it as supplementary material.

BR,

Sebastian Springer

---

## Referee Comment (RC1) · Anonymous Referee #1 · 21 Dec 2020

\*\*\* General comments \*\*\*

This paper combines and applies two other previously published methods to enable robust Bayesian inference of parameters for chaotic dynamical systems via Markov chain Monte Carlo (MCMC): the correlation integral likelihood (CIL) makes probabilistic inference on chaotic systems reliable, while local approximation MCMC (LA-MCMC) makes it efficient. This opens up MCMC sampling methods to work on numerically expensive chaotic models such as climate models, for which they would otherwise remain intractable.

The use of emulators as stand-ins for computationally expensive models under MCMC is a well-established area. The use of Bayesian techniques on chaotic systems seems to be less well trod, but potentially very high impact as our understanding of these

systems is pivotal to our understanding of the environment and to decision making for resource management and environmental stewardship. The correlation integral likelihood has received relatively little attention compared to synthetic likelihood approaches such as Wood 2010, but the computational efficiency gains for this type of system are clearly demonstrated. The value of the paper thus lies in a proof of principle for applying these techniques together to solve more challenging problems. The CIL might be enough on its own to handle the Lorentz attractor and simple population dynamics models, and LA-MCMC has been demonstrated on inversions involving partial differential equations. For complex nonlinear PDEs such as those describing weather systems, both are probably needed.

Since this paper presents applications rather than all-new methods, it makes sense to submit to an earth science journal rather than an applied statistics journal. Chaotic systems such as weather and climate models are within scope for GMD, and Bayesian inference could be viewed as a means for comparing models to data. The results are general and don't focus on any single geoscientific forward model, though the CIL is a likelihood and thus is an important part of a *statistical* model for the data. If all the forward models were coded with a specific library or package, or if the complete method were coded in a single forward-model-agnostic library, it would be natural to include the code name and version number in the title, but that doesn't seem to be the case; I'll defer to the editor's judgment in this matter.

The paper is overall well-structured and clearly and concisely demonstrates the practical application of the methodology. Model setups are clearly described. The first two experiments are useful to demonstrate that LA-MCMC recovers the true posterior for chaotic systems using CIL, and the reduction of two years' worth of conventional MCMC to three hours' walltime with the quasi-geostrophic weather model is impressive. The description of CIL is detailed and would enable anyone to code the metric for themselves, and the authors cite a good representative sample of competing methods for CIL, and convincingly demonstrate that their method gives sensible results.

[Figure]

I found the paper interesting and informative and believe it merits publication. The main class of clarifications I would like to see concern how various choices were made for the setup and parameters of the CIL and LA-MCMC algorithms used to attack these problems, which will be of interest to any practitioners who want to adopt this method for their problems.

*** Specific comments ***

– Section 2.2 (Local Approximation MCMC)

The discussion of LA-MCMC is shorter and more schematic than the discussion of CIL in the preceding subsection, and relies heavily on citations of Davis et al 2020. Repeating the entire algorithm setup may be overkill, but this level of description is probably not enough for the user to understand and reproduce how the algorithm works, in contrast to the CIL section. There is some discussion about the benefits of using emulators in general, where I think it's appropriate to cite some key papers (e.g. Sacks et al 1989; Kennedy & O'Hagan 2001) introducing the idea of emulators for model calibration and efficiency in Bayesian inference. The specific advantage of LA-MCMC as a meta-method – not the fact of being an emulator method, nor its functional form, but the theoretical guarantee of optimal convergence as the number of samples and forward model evaluations increases – is reasonably well made here. Some discussion about the local support of the approximation and refinement criteria, as in Conrad et al 2016, would also be welcome in this subsection.

Why were the particular parameters to LA-MCMC chosen here? Were there any problem-specific considerations, or are these reasonable default settings? For example, was the quadratic approximation chosen for any particular reason over either linear or Gaussian-Process-based local approximations? How would different choices impact the performance?

What Metropolis-Hastings proposal density is actually used for these problems and how was it tuned? The algorithm as outlined in Davis et al 2020 is a meta-method

that specifies only that the proposal density should be constant in time, which seems on its face to rule out many adaptive and gradient-based proposals. The number of parameters being inferred is always small in this paper, so a Gaussian random walk might work fine, but if so please state that.

– Section 3 (Numerical Experiments)

What considerations are used to pick the number M of correlation scale bins? Clear criteria are given for R_0 and R_M, and b is fixed once these parameters and M are known. The specific choices of M for each problem are given in the text, but not why they are optimal or even appropriate. How will performance of the CIL vary if M is suboptimal?

Similarly, how is the observation interval Delta_t chosen? It seems in all application cases to be deliberately longer than the "predictable" interval, presumably to ensure that adjacent observations yield new information and have settled into the attractor geometry. This becomes important specifically when simulating many short chaotic trajectories rather than one long one (to parallelize the problem and achieve desired efficiency gains). These points are alluded to in the subsequent subsections for individual experiments, but they are central enough points to bring up or reiterate alongside other general information in the opening part of the section.

This would also be a good place to talk about the computing architecture used for these experiments, since it's hard to see what level of computational effort this wall time signifies otherwise. (Mention of a NVIDIA 1070 GPU is made once, and only for the K-S model, though comments about the degree of parallelization are mentioned for each problem.) This is more for completeness given that the choice of processor won't make a factor of 100 difference to the results, but practitioners may want to know this. If any particular libraries were used e.g. fast solvers or integrators, those dependencies should also be made clear.

– Section 3.2 (Quasi-geostrophic model)

[Figure]

For readers unfamiliar with this model, are H1 and H2 only mean thicknesses, or is this more of a 2.5-dimensional model where H1 and H2 are spatially constant thicknesses of two interacting vertical zones?

What was the value of f0 used? How do the beta coefficients depend upon it (dependencies of all other parameters on f0 are given in the text)? Was f0 inferred alongside H1 and H2? If not, would sampling over it be just as straightforward as sampling over H1 and H2, or would it be likely to create new challenges due to its correlations with several other parameters of the problem – as for example sampling in hierarchical Bayesian models where a global variance scale can induce strong and inconvenient posterior curvature?

– Section 4 (Conclusions)

Are there any specific inference problems currently faced by the weather and climate modeling community that might not have been previously feasible, but could perhaps now be attempted with this method?

– Supplementary Material

The authors have expressed their wish to include the code for these examples as supplementary material. Ideally, as a reader I'd prefer to see a Git repository where the code can be downloaded and installed at will, tagged with a given version, and given an external persistent DOI, even if it is not meant to be maintained as a general open-source package for the community. If the authors are going to upload the code as a supplement, it becomes much more important to clearly indicate all environment parameters and code dependencies, and to provide the supplement in a form that can simply be downloaded into a directory and run by the user to reproduce their key results.

\*\*\* Technical comments \*\*\*

Fig 1: The layout of this figure could be expanded to make it more readable. The font

for the labels seems quite small, and the vertical arrangement of the graphs is crowded. All content is appropriate, though.

Figs 2, 4, 6, 8, 9: Similarly expand label fonts a bit for easier reading, and adjust tick spacing if necessary to prevent overlap.

Fig 3: There isn't much information in this figure and I believe it could be safely omitted. Using a linear y-axis makes the effort needed for full MCMC look dramatic compared to LA-MCMC, but this point is adequately made in the text. The figure would make more sense if the paper were about comparing the efficiencies of different emulator methods.

Fig 7: This is a useful cartoon; I'd like to see some of the other geometric parameters of the system included on this figure, if possible, given that the actual version solved becomes non-dimensional.

---

## Referee Comment (RC2) · Anonymous Referee #2 · 31 Dec 2020

General comments

This paper aims to solve the high computational challenge in the statistical inference of chaotic dynamic models. This is a practical and challenging problem. It has great potential in practice. This paper combines two methods to solve the intractability of inference of chaotic dynamic models. CIL was adopted to incorporate more information of the observations into the summary statistics. LA-MCMC was utilized to reduce computational time and thus make the proposed method more practical.

The paper was written in a very good manner. First of all, the problem was introduced in Section 1. Then CIL and LA-MCMC were described in detail in Section 2. Three examples were demonstrated in Section 3. The authors have put a lot of effort to CIL and LA-MCMC, especially CIL. This makes the paper self-contained. To improve the

**GMDD**

readability, the authors should state clearly in Section 2 which parts are novel. This helps readers to understand the contribution of this work.

It seems this paper does not match well with the scope of Geoscientific Model Development (Methods for assessment of models). It should be submitted to an applied statistical journal. This paper described the state-of-art statistical methods and implemented several experiments to evaluate the performance of such methods in the applications of geoscience. The paper provided codes for practitioners to run their own data. The paper itself is rich in information that is very useful for practitioners, but it may not be appropriate to publish in GMD journal. According to the scope of GMD journal, the publication should develop new metrics for assessing model performance and novel ways of comparing model results with observational data. I cannot see such work in this paper. It also worries me that the authors made many decisions without enough support either from theory or empirical evidence. For example, in line 201, the authors claim that their likelihood function does not depend on the initial conditions of the forward model. Is this statement true only to your forward model? In general, the paper merits publication given the following comments can be responded properly.

Specific comments

1. The statistical properties of CIL are crucial to determine the overall performance of the proposed method. Does CIL converge to the true likelihood function and in what conditions it converges to the true likelihood? In approximate Bayesian computation, the approximate likelihood will converge to the true likelihood function given that the summary statistics are sufficient and the threshold approaches to zero. What conditions are essential for CIL to converge?

2. It is not clear what novelty of this paper is. Both CIL and LA-MCMC are well developed methods. What is the contribution of this paper? In section 2, the authors basically reviewed two methods: CIL and LA-MCMC. This paper does not propose new geoscience models either. It is not clear which parts are proposed by the au-

thors and what novelty is. Please state in the paper clearly which parts are new to the literature, either in methodological or domain area.

3. Line 201: the authors claim that their likelihood function does not depend on the initial conditions of the forward model. This is a very ambitious declaration. From my understanding, the initial conditions of the forward model will impact the synthetic data and thus impact the summary statistics significantly. Please explain why your statement is true and show some evidence.

4. Line 206: the authors stated their approach can save computational time by reducing the length of simulated forward model. They only require one single epoch to compute the CIL for the later inference. The idea is beneficial to save computational resources. However, this may lead to skewed posterior distributions. The reason is that the mean vector and covariance matrix in Equation (4) are computed based on all the combinations of s and l. Normally, a synthetic data should be of the same length as the observation and Equation (5) can be computed correspondingly. The current version of Equation (5) is likely to lead to a skewed posterior distribution, because only partial comparison between the synthetic data and the observations has been incorporated into the likelihood function. Intuitively, the Figure 2 and Figure 4 have shown some skewness in the posterior distributions. Can you explain and justify your reasonings behind the line 206?

5. As the data set in the simulation of Lorenz 63 model, we should expect LA-MCMC to matches standard MCMC very well. Figure 2 demonstrates the pairwise marginal distribution of Lorenz 63 model. Does Lorenz 63 model pay more attention to the accuracy of pairwise marginal distributions? Otherwise, the authors should show the results of each single parameters, so the readers can evaluate the performance more easily. For the other models, can you show the marginal distributions of each single parameter as well?

---

## Author Response (AR1)

**gmd-2020-350: responses to reviewer comments**

Sebastian Springer, Heikki Haario, Jouni Susiluoto, Aleksandr Bibov, Andrew Davis, and Youssef Marzouk

We thank the anonymous reviewers and the editor for carefully reading the manuscript and for providing the very valuable comments. We address the comments one by one below. The reviewer comments are pasted verbatim below in italics, and the author responses to these comments can be found immediately under the comments, starting "A:". These are followed by "**Changes to manuscript:**" sections, where the line numbers refer to the diff file unless stated otherwise. Line numbers in these "A:" sections generally refer to the old version of the manuscript.

**Anonymous Referee #1**

This paper combines and applies two other previously published methods to enable robust Bayesian inference of parameters for chaotic dynamical systems via Markov chain Monte Carlo (MCMC): the correlation integral likelihood (CIL) makes probabilistic inference on chaotic systems reliable, while local approximation MCMC (LA-MCMC) makes it efficient. This opens up MCMC sampling methods to work on numerically expensive chaotic models such as climate models, for which they would otherwise remain intractable.

The use of emulators as stand-ins for computationally expensive models under MCMC is a well-established area. The use of Bayesian techniques on chaotic systems seems to be less well trod, but potentially very high impact as our understanding of these systems is pivotal to our understanding of the environment and to decision making for resource management and environmental stewardship. The correlation integral likelihood has received relatively little attention compared to synthetic likelihood approaches such as Wood 2010, but the computational efficiency gains for this type of system are clearly demonstrated. The value of the paper thus lies in a proof of principle for applying these techniques together to solve more challenging problems. The CIL might be enough on its own to handle the Lorentz attractor and simple population dynamics models, and LA-MCMC has been demonstrated on inversions involving partial differential equations. For complex nonlinear PDEs such as those describing weather systems, both are probably needed.

Since this paper presents applications rather than all-new methods, it makes sense to submit to an earth science journal rather than an applied statistics journal. Chaotic systems such as weather and climate models are within scope for GMD, and Bayesian inference could be viewed as a means for comparing models to data. The results are general and don't focus on any single geoscientific forward model, though the CIL is a likelihood and thus is an important part of a \*statistical\* model for the data. If all the forward models were coded with a specific library or package, or if the complete method were coded in a single forward-model-agnostic library, it would be natural to include the code name and version number in the title, but that doesn't seem to be the case; I'll defer to the editor's judgment in this matter.

The paper is overall well-structured and clearly and concisely demonstrates the practical application of the methodology. Model setups are clearly described. The first two experiments are useful to demonstrate that LA-MCMC recovers the true posterior for chaotic systems using CIL, and the reduction of two years' worth of conventional MCMC to three hours' walltime with the quasi-geostrophic weather model is impressive. The description of CIL is detailed and would enable anyone to code the metric for themselves, and the authors cite a good representative sample of competing methods for CIL, and convincingly demonstrate that their method gives sensible results.

I found the paper interesting and informative and believe it merits publication. The main class of clarifications I would like to see concern how various choices were made for the setup and parameters of the CIL and LA-MCMC algorithms used to attack these problems, which will be of interest to any practitioners who want to adopt this method for their problems.

A: We thank the reviewer for this sincere assessment.

1. The discussion of LA-MCMC is shorter and more schematic than the discussion of CIL in the preceding subsection, and relies heavily on citations of Davis et al 2020. Repeating the entire algorithm setup may be overkill, but this level of description is probably not enough for the user to understand and reproduce how the algorithm works, in contrast to the CIL section. There is some discussion about the benefits of using emulators in general, where I think it's appropriate to cite some key papers (e.g. Sacks et al 1989; Kennedy & O'Hagan 2001) introducing the idea of emulators for model calibration and efficiency in Bayesian inference. The specific advantage of LA-MCMC as a meta-method – not the fact of being an emulator method, nor its functional form, but the theoretical guarantee of optimal convergence as the number of samples and forward model evaluations increases – is reasonably well made here. Some discussion about the local support of the approximation and refinement criteria, as in Conrad et al 2016, would also be welcome in this subsection.

A1: A more detailed description has been added (p.12, l. 280-281 and 288-289) to help the user to reproduce the LA-MCMC method. In addition, a commented Matlab code is available for the user, that contains the main steps and, naturally, all the constants and other details in the subroutines. References to the mentioned key emulator papers have been added in the text (p.10, l. 223-224). As for the local support and refinement criteria, we note the at a given point, we locally compute a polynomial approximation using the nearest neighbors in the evaluated set as data for regression. However, we emphasize that the approximation itself is not locally supported—it is a piecewise polynomial that is defined over all of parameter space. This is an important distinction because the piecewise polynomial approximation to the posterior density function (equivalently, the likelihood density function) is not necessarily itself a probability density function. In fact, the surrogate function may not even be integrable. Despite this challenge, Davis et al. 2020 devise a refinement strategy that ensures convergence and bounds the error after a *finite* number of samples. This strategy uses an estimate of the local bound on the surrogate error to trigger local

refinement. We employ a similar strategy in this paper. Discussion on this has been added in the text.

Changes to manuscript: As given in the reply.

2. Why were the particular parameters to LA-MCMC chosen here? Were there any problemspecific considerations, or are these reasonable default settings? For example, was the quadratic approximation chosen for any particular reason over either linear or Gaussian-Process-based local approximations? How would different choices impact the performance?

A2: At the end of section 2.2 we include some specific parameter choices. The same values are used in all the examples, so they may be regarded as reasonable default settings here. However, in principle the parameters can be problem specific. Davis et al. provides some intuition for parameter choices. The primary tuning parameter is the initial error threshold—the decay rate is then set by LA-MCMC to ensure the *rates* are optimal. We specifically choose a quadratic model because in the Gaussian case this leads to an exact approximation (the approximation is the log-target, which is a quadratic in the Gaussian case). All the specific parameter choices are also available in the supplementary Matlab code, and may be freely tuned by the user.

**Changes to manuscript:** The parameter choices have been further discussed at the end of Sect. 2.2, together with a note pointing to the MATLAB code in the code availability section.

3. What Metropolis-Hastings proposal density is actually used for these problems and how was it tuned? The algorithm as outlined in Davis et al 2020 is a meta-method that specifies only that the proposal density should be constant in time, which seems on its face to rule out many adaptive and gradient-based proposals. The number of parameters being inferred is always small in this paper, so a Gaussian random walk might work fine, but if so please state that.

A3: The theory in Davis et al. 2020 indeed assumes a constant-in-time proposal density. However, this does not necessarily imply that adaptive or gradient-based methods will not converge. In particular, Conrad et al. 2018 show that the Metropolis adjusted Langevin algorithm, which is a gradient based MCMC method, is asymptotically exact when using a continually refined local polynomial approximation. This proof requires some additional assumptions about the target density's tail behavior, and the stronger rate-optimal result from Davis et al. has not yet been established for such algorithms. In practice, however, we see that adaptive methods still work well in our applications. Exploring the theoretical implications of this is interesting and merits further discussion but is beyond the scope of this paper. A standard random walk Metropolis sampler with a fixed Gaussian proposal works fine, in case a good proposal has been found.

**Changes to manuscript:** We now mention that a range of MCMC methods, including standard Gaussian random walk, can be used with LA-MCMC. The discussion has been added in Sect. 2.2, last paragraph on p. 11

4. What considerations are used to pick the number M of correlation scale bins? Clear criteria are given for  $R_0$  and  $R_M$ , and b is fixed once these parameters and M are known. The specific choices of M for each problem are given in the text, but not why they are optimal or even appropriate. How will performance of the CIL vary if M is suboptimal? Similarly, how is the

observation interval  $\Delta_t$  chosen? It seems in all application cases to be deliberately longer than the "predictable" interval, presumably to ensure that adjacent observations yield new information and have settled into the attractor geometry. This becomes important specifically when simulating many short chaotic trajectories rather than one long one (to parallelize the problem and achieve desired efficiency gains). These points are alluded to in the subsequent subsections for individual experiments, but they are central enough points to bring up or reiterate alongside other general information in the opening part of the section. This would also be a good place to talk about the computing architecture used for these experiments, since it's hard to see what level of computational effort this wall time signifies otherwise. (Mention of a NVIDIA 1070 GPU is made once, and only for the K-S model, though comments about the degree of parallelization are mentioned for each problem.) This is more for completeness given that the choice of processor won't make a factor of 100 difference to the results, but practitioners may want to know this. If any particular libraries were used e.g. fast solvers or integrators, those dependencies should also be made clear.

A4: As always with histograms, the selection of the number M of bins is a bit problematic. Too small M loses information, while too large M yields 'noisy' histograms and CDFs. However, numerical experiments show that the final result, the parameter posteriors, are not too sensitive to the specific value of M. For instance, for the 3D Lorenz case the range of M was varied between 5 and 40, and only a minor decrease of the size of the parameter posteriors was noticed for increasing M. However, the slight increase of accuracy comes with a CPU cost: high values of M increase the stochasticity of the likelihood evaluations, that leads to small acceptance ratios in the MCMC sampling: from about 0.36 to 0.17 to 0.03 for M = 5,15,40, respectively, when using standard AM. The trend is the same with LA-MCMC, but the use of LA-MCMC removes some of the stochasticity and typically leads to higher acceptance ratios. We admit that the choice of M needs some hand-tuning but this has not been any major issue in any of the cases studied. The choice of  $\Delta_t$  was indeed taken large, beyond the threshold of predictability. This was more for demonstration purposes than necessity; the background theory from U-statistics allows that the subsequent state vectors are weakly dependent. Numerically, a too high density of observation points is revealed by a failure of the  $\chi^2$  test, that always is recommended to be performed before starting the parameter estimation.

The KS-model is based on our in-house FFT-based solver, which runs on the GPU-side and is built around NVIDIA CUDA toolchain and cuFFT library (which is a part of the CUDA ecosystem). The Quasi-Geostrophic model employs semi-Lagrangian solver and runs entirely on CPU, but the code has been significantly optimized with performance-critical parts, such as advection operator, compiled using ISPC with support of AVX2 vectorization.

**Changes to manuscript:** The M insensitivity comment is copied to the main text. A comment on the choice of  $\Delta_t$  has been added. (p.13-14, l. 305-310 and 326-334), and the computing architecture at the end of the section.

5. For readers unfamiliar with this model, are H1 and H2 only mean thicknesses, or is this more of a 2.5-dimensional model where H1 and H2 are spatially constant thicknesses of two interacting vertical zones? What was the value of f0 used? How do the beta coefficients depend upon it (dependencies of all other parameters on f0 are given in the text)? Was f0 inferred alongside H1 and H2? If not, would sampling over it be just as straightforward as sampling over H1 and H2, or would it be likely to create new challenges due to its correlations with several other parameters of the problem – as for example sampling in hierarchical Bayesian models where a global variance

**scale can induce strong and inconvenient posterior curvature?**

A5: H1 and H2 denote the mean thicknesses of the layers and in the quasi-geostrophic model these layers physically interact. The formula for f0, now given in the text, shows how f0 is a known expression of well-known factors (the latitude and the rotation speed of the earth) so f0 might not be a natural parameter to be sampled, even for demonstration purposes. Situations with correlated and higher dimensional parameters are covered in the other examples. A three dimensional parameter vector is considered in the Lorenz example. The KS example provides a case with a strongly correlated posterior. We agree that other parameters (than f0, e.g., those giving the orography) might be sampled here but we do not expect that any major issues would appear in the QG case. Models with higher dimensional unknowns are considered in the references given in Section 2.2 line 240.

**Changes to manuscript:** The role of the parameter f0 has been clarified. (p.21-22, l. 455-459) We expand the description of H1 and H2 above Eq. (13) and (14) to explain that they are mean thicknesses of interacting layers.

**6. Are there any specific inference problems currently faced by the weather and climate modeling community that might not have been previously feasible, but could perhaps now be attempted with this method?**

A6: The present work gives a solution to the problem discussed in the Introduction concerning the estimation of climate model parameters. In Järvinen et al. (2010) a study was performed for the ECHAM6 model parameters. However, the naive summary statistics approach used there were not able to identify the parameters studied, even if the MCMC sampling was technically successfully performed. We believe that the CIL is much more sensitive with respect to climate model parameters and will help to calibrate such models. Also, the discussion in Rougier (2013) points out the intractable character of problems, which we believe are now coming feasible by our approach. These points were already discussed in the Introduction, but we now explicitly return to them in the Conclusion.

Changes to manuscript: This item is underlined in the Conclusion. (p.27, l. 514-517)

7. The authors have expressed their wish to include the code for these examples as supplementary material. Ideally, as a reader I'd prefer to see a Git repository where the code can be downloaded and installed at will, tagged with a given version, and given an external persistent DOI, even if it is not meant to be maintained as a general open source package for the community. If the authors are going to upload the code as a supplement, it becomes much more important to clearly indicate all environment parameters and code dependencies, and to provide the supplement in a form that can simply be downloaded into a directory and run by the user to reproduce their key results.

A7: We appreciate the suggestion to use a public Git and persistent DOI. We will make the code available on Github in this way, and/or on any other platform that the journal prefers. **Changes to manuscript:** Code for the general algorithms as well as each numerical example (L3, KS, QG) will be made available and links to the relevant repositories will be given in the code availability section.

8. Fig 1: The layout of this figure could be expanded to make it more readable. The font for the labels seems quite small, and the vertical arrangement of the graphs is crowded. All content is appropriate, though. Figs 2, 4, 6, 8, 9: Similarly expand label fonts a bit for easier reading, and adjust tick spacing if necessary to prevent overlap. Fig 3: There isn't much information in this figure and I believe it could be safely omitted. Using a linear y-axis makes the effort needed for full MCMC look dramatic compared to LA-MCMC, but this point is adequately made in the text. The figure would make more sense if the paper were about comparing the efficiencies of different emulator methods. Fig 7: This is a useful cartoon; I'd like to see some of the other geometric parameters of the system included on this figure, if possible, given that the actual version solved becomes non-dimensional.

A8: The labels have been corrected to be larger. Fig 3 is perhaps obvious, given the surrounding discussion in the text, but it communicates a central point of the paper.

Changes to manuscript: The figures are corrected.

**Anonymous Referee #2**

This paper aims to solve the high computational challenge in the statistical inference of chaotic dynamic models. This is a practical and challenging problem. It has great potential in practice. This paper combines two methods to solve the intractability of inference of chaotic dynamic models. CIL was adopted to incorporate more information of the observations into the summary statistics. LA-MCMC was utilized to reduce computational time and thus make the proposed method more practical.

The paper was written in a very good manner. First of all, the problem was introduced in Section 1. Then CIL and LA-MCMC were described in detail in Section 2. Three examples were demonstrated in Section 3. The authors have put a lot of effort to CIL and LA-MCMC, especially CIL. This makes the paper self-contained. To improve the readability, the authors should state clearly in Section 2 which parts are novel. This helps readers to understand the contribution of this work.

It seems this paper does not match well with the scope of Geoscientific Model Development (Methods for assessment of models). It should be submitted to an applied statistical journal. This paper described the state-of-art statistical methods and implemented several experiments to evaluate the performance of such methods in the applications of geoscience. The paper provided codes for practitioners to run their own data. The paper itself is rich in information that is very useful for practitioners, but it may not be appropriate to publish in GMD journal. According to the scope of GMD journal, the publication should develop new metrics for assessing model performance and novel ways of comparing model results with observational data. I cannot see such work in this paper. It also worries me that the authors made many decisions without enough support either from theory or empirical evidence. For example, in line 201, the authors claim that their likelihood function does not depend on the initial conditions of the forward model. Is this statement true only to your forward model? In general, the paper merits publication given the following comments can be responded properly.

A: We thank the reviewer for this sincere assessment. We understand the worry about the suitability of the topic to GMD, but would like to point out that chaotic dynamical systems appear in nature frequently, and that the models used to test the LA-MCMC+CIL method are used frequently in geophysical studies. We believe that the comments made by anonymous referee #1 are spot-on regarding the suitability of this paper to GMD.

Probabilistic model calibration is a very useful ingredient in model development and validation. Since the original algorithms have been published elsewhere, we don't believe a statistics journal would be the correct forum. The novelty of the work is to show how the combination of the methods – as the reviewer correctly points out above – enables one to solve problems hitherto considered intractable. The focus of this paper is to showcase geoscientific toy applications to demonstrate how CIL and MCMC can be combined to tackle such questions. We believe this is in the interest of practitioners reading GMD.

The CIL approach can be used with both simulated model data and real measurements, and especially in cases where comparisons between data and model have been difficult or intractable so far. Therefore the paper directly relates to "assessing model performance and novel ways of comparing model results with observational data."

**Changes to manuscript:** We have tried to make this point with more clarity in the introduction and discussion of the paper.

9. The statistical properties of CIL are crucial to determine the overall performance of the proposed method. Does CIL converge to the true likelihood function and in what conditions it converges to the true likelihood? In approximate Bayesian computation, the approximate likelihood will converge to the true likelihood function given that the summary statistics are sufficient and the threshold approaches to zero. What conditions are essential for CIL to converge?

A9: As for the theoretical background, we actually do refer to literature regarding the normality of the statistic, see I. 170. We would like to point out that the CIL *is* a Gaussian likelihood, and that the results have been shown to follow Gaussian statistics. Convergence in the Bayesian context to the appropriate posterior follows from this observation. What is approximate here, however, is the estimation of the mean and covariance of the Gaussian likelihood. The estimates do converge with more data (with more epochs), the rate of convergence is  $1/\sqrt{n_{epo}}$ , as shown in the U-statistics literature (references listed in Section 2.1, line 173). LA-MCMC estimates become exact when chain lengths go to infinity.

Changes to manuscript: We add a comment about the convergence at (p.8, l. 174-175).

10. It is not clear what novelty of this paper is. Both CIL and LA-MCMC are well developed methods. What is the contribution of this paper? In section 2, the authors basically reviewed two methods: CIL and LA-MCMC. This paper does not propose new geoscience models either. It is not clear which parts are proposed by the authors and what novelty is. Please state in the paper clearly which parts are new to the literature, either in methodological or domain area.

A10: The main novelty for readers with geosciences backgrounds is demonstrating how model parameters can be identified in large and computationally heavy chaotic systems. This has not been done before (see Järvinen et al. (2010), Rougier, (2013)). This requires both the CIL and LA-MCMC methods, that together enable, for instance, to reduce the computational time of 2 years to 4 houres in the QG example. Additionally, 'technical' novelties are employed in the paper that are not published before: parallel calculations by splitting the time-integration in parts, and the use of multiple likelihood evaluations together with ridge regression to reduce the noise in the surrogate regression surface construction in the LA-MCMC calculations.

**Changes to manuscript:** The novelty for the GMD readers is more clearly underlined in the Introduction and Conclusion. (p.4, l. 73-80 and p.27 l. 514-517)

11. Line 201: the authors claim that their likelihood function does not depend on the initial conditions of the forward model. This is a very ambitious declaration. From my understanding, the initial conditions of the forward model will impact the synthetic data and thus impact the summary statistics significantly. Please explain why your statement is true and show some evidence.

A11: Chaotic models need to be integrated long enough so that the initial state becomes irrelevant. It is a common practice used in, e.g., atmospheric climate modelling to drop away a few initial months for this purpose, and also to ensure that the computed state values are on the underlying attractor of the system. In our approach the initial values are randomized for all simulations, and samples are taken only after the predictable initial time interval. Moreover, the independency of the sampled parameter posteriors from the initial values has been extensively verified both here and in earlier works by repeated numerical experiments. We agree with the reviewer that one has to be careful here, however. With chaotic models it's possible that there are rare events or regions of the attractor that are not represented in the data at all. With models with very slow dynamics multiple random initializations may be used to make sure that the sampling takes place as widely as possible in the attractor. For the models presented in this work, we believe that the way the integration is performed is sufficiently long so that we can claim that the model "forgets" the initial value and covers the relevant parts of the attractor.

**Changes to manuscript:** We have added a note about these points in the text. (p.9, l. 207-210).

12. Line 206: the authors stated their approach can save computational time by reducing the length of simulated forward model. They only require one single epoch to compute the CIL for the later inference. The idea is beneficial to save computational resources. However, this may lead to skewed posterior distributions. The reason is that the mean vector and covariance matrix in Equation (4) are computed based on all the combinations of s and l. Normally, a synthetic data should be of the same length as the observation and Equation (5) can be computed correspondingly. The current version of Equation (5) is likely to lead to a skewed posterior distribution, because only partial comparison between the synthetic data and the observations has been incorporated into the likelihood function. Intuitively, the Figure 2 and Figure 4 have shown some skewness in the posterior distributions. Can you explain and justify your reasonings behind the line 206?

A12: In the training stage the likelihood is indeed constructed for one epoch only, by repeatedly

using data from all available epochs. In the MCMC sampling stage, the very same calculation is performed, now by computing the distances between the proposed new one-epoch trajectory and one trajectory from the training data set. But we also can use several trajectories from the training data. In Eq. 5 we do this by averaging the calculation over several training data trajectories. This is beneficial as it reduces the stochasticity of the likelihood evaluation. Naturally, the likelihood construction in the training stage and the evaluations of it in the sampling stage must again be performed in exactly the same way.

As for Figures 2 and 4, they do not show any systematic skewness. The 'true' parameter should (as always in MCMC sampling) lie inside the posterior, but not necessarily close to the expected value. With new experiments with new data sets the posterior moves around the 'true' parameter, but without any systematic skewness. Even with the same data and using just the standard AM sampling the posteriors slightly vary if the sampling is repeated. The difference between the posteriors obtained by AM and LA-MCMC sampling is approximatively at that level.

**Changes to manuscript:** A comment concerning the above point has been added in the text. (p.15, l. 371-372)

13. As the data set in the simulation of Lorenz 63 model, we should expect LA-MCMC to matches standard MCMC very well. Figure 2 demonstrates the pairwise marginal distribution of Lorenz 63 model. Does Lorenz 63 model pay more attention to the accuracy of pairwise marginal distributions? Otherwise, the authors should show the results of each single parameters, so the readers can evaluate the performance more easily. For the other models, can you show the marginal distributions of each single parameter as well?

A13: Reporting the pairwise marginal distributions is done in the paper for visualization purposes only, since visualization of higher dimensional target distributions is challenging. The pairwise marginal distributions, unlike univariate marginal distributions, still retain information about correlations of the samples in the posterior, which can be useful for the reader. We point out that showing agreement of 2-d marginal distributions is a stricter criterion than just demonstrating agreement of univariate marginal distributions. The results of MCMC posteriors naturally change for any new simulations and data cases, see above the reply to the previous question.

Figure 2 has been edited as suggested by the reviewer. The single parameter distributions are added to the other models (Figures 4 and 9 - except left away, for clarity, from Fig 6 that contains several posteriors corresponding to different amounts of data).

Changes to manuscript: We have changed Figures 2, 4 and 9.

---

## Author Response (AR2)

**gmd-2020-350: responses to reviewer comments**

Sebastian Springer, Heikki Haario, Jouni Susiluoto, Aleksandr Bibov, Andrew Davis, and Youssef Marzouk

We thank the anonymous reviewers and the editor for carefully reading the manuscript and for providing valuable comments. We address the comments one by one below. The reviewer comments are pasted verbatim below in italics, and the author responses to these comments can be found immediately under the comments, starting "A:". These are followed by "**Changes to manuscript:**" sections. All three edits are highlighted with boldface in the text of the article, all in Section 2 there.

**Anonymous Referee #1**

The authors have addressed most of my earlier suggestions and I find the paper reads better as a result. I have a few more minor suggestions:

(1) While I understand that a more concrete summary of algorithmic steps in the LA-MCMC algorithm may be too far afield for this paper, I wouldn't assume that all readers are familiar with Matlab or will have access to Matlab licenses. Some specific words like "we implement Algorithm 1 from Davis et al 2020, with the following parameter values as defined in that paper", would help the readers to know exactly where to go in that paper to see what is being done.

A1: **Changes to manuscript:** The respective sentence has been edited, and the reference to Davis et al 2020 has been added.

(2) If the authors used ridge regression in their implementation of Davis et al 2020, I assume there is a hyperparameter associated with the strength of the L2 penalty on polynomial coefficients. I don't see such a hyperparameter mentioned in the text, nor comments about how they would have set its value – though it could be done through cross-validation which is already performed on the local regression results as the algorithm runs. Anyway, it would be good to see some brief mention of this, especially since it is a variation on Davis et al 2020.

A2: In numerical experiments it turned out that the results of the ridge regression were rather insensitive with respect to the spesific value of the regularization/hyperparameter  $\alpha$  used, as long as it was roughly in the range (0.1,10) (similarly as is shown, e.g., in the demo of the Matlab's ridge function help). The choice  $\alpha = 1$  was then selected to be used.

**Changes to manuscript:** The regression has been explained with a bit more details in the text, and the hyperparameter value is given.

(3) The mention of previous classic emulator papers should actually include citations to those papers in the bibliography! The references are Sacks, J., W. J. Welch, T. J. Mitchell, and H. P. Wynn (1989). Design and analysis of computer experiments. Statistical Science 4(4), 409–423. Kennedy, M. and A. O'Hagan (2001). Bayesian calibration of computer models. Journal of the Royal Statistical Society: Series B (Statistical Methodology) 63(3), 425–464.

A3:

Changes to manuscript: The references have been added.

---

## Author Response (AR3)

We thank the editor for the constructive comments. The comments are addressed below under the comments, which are pasted in *italics*.

**Topical Editor Decision: Publish subject to minor revisions (review by editor)** (07 May 2021) by Rohitash Chandra
Comments to the Author:
*The manuscript needs to have a Background and related work section to provide more details of what has been done in the literature. The literature review should extend and include recent works in the area of MCMC and the related problems of interest in this paper.*

We have split the Introduction into two sections: "1. Introduction", and "2. Background and related work" as requested. We have expanded the literature review and added new references, including to relevant MCMC topics.

*The literature review is not adequate and also the experiments do not show results compared with published methods from the literature.*

We are not aware of any literature where parameters of chaotic models were estimated using observations that are temporally distant from each other. We have expanded discussion of filtering and summary statistics-based methods in sections 1 and 2.

*Need a table of results summary and if any related works are there in the literature - include them for comparison.*

See previous answer above. We have added a table summarizing the speed-ups provided by the LA-MCMC method in the Conclusions.